# Multiple Roles of Chitosan in Mucosal Drug Delivery: An Updated Review

**DOI:** 10.3390/md20050335

**Published:** 2022-05-20

**Authors:** Paola Mura, Francesca Maestrelli, Marzia Cirri, Natascia Mennini

**Affiliations:** Department of Chemistry, University of Florence, Via Schiff 6, Sesto Fiorentino, 50019 Florence, Italy; francesca.maestrelli@unifi.it (F.M.); marzia.cirri@unifi.it (M.C.); natascia.mennini@unifi.it (N.M.)

**Keywords:** chitosan, buccal delivery, nasal delivery, ocular delivery, rectal delivery, vaginal delivery, oral delivery

## Abstract

Chitosan (CS) is a linear polysaccharide obtained by the deacetylation of chitin, which, after cellulose, is the second biopolymer most abundant in nature, being the primary component of the exoskeleton of crustaceans and insects. Since joining the pharmaceutical field, in the early 1990s, CS attracted great interest, which has constantly increased over the years, due to its several beneficial and favorable features, including large availability, biocompatibility, biodegradability, non-toxicity, simplicity of chemical modifications, mucoadhesion and permeation enhancer power, joined to its capability of forming films, hydrogels and micro- and nanoparticles. Moreover, its cationic character, which renders it unique among biodegradable polymers, is responsible for the ability of CS to strongly interact with different types of molecules and for its intrinsic antimicrobial, anti-inflammatory and hemostatic activities. However, its pH-dependent solubility and susceptibility to ions presence may represent serious drawbacks and require suitable strategies to be overcome. Presently, CS and its derivatives are widely investigated for a great variety of pharmaceutical applications, particularly in drug delivery. Among the alternative routes to overcome the problems related to the classic oral drug administration, the mucosal route is becoming the favorite non-invasive delivery pathway. This review aims to provide an updated overview of the applications of CS and its derivatives in novel formulations intended for different methods of mucosal drug delivery.

## 1. Introduction

Oral drug administration, although remaining the most widely used route and the favorite of patients, mainly due to its safety in use and easy, painless and noninvasive self-administration, presents some important drawbacks, including drug instability in the gastric acidic environment, enzymatic degradation during the gastrointestinal transit and hepatic first-pass metabolism, that preclude its use for certain classes of drugs, including peptides and proteins. Moreover, unsatisfactory drug targeting to the site of action often requires higher drug concentrations, thus increasing the risk of adverse side effects.

Over the past few decades, considerable research into novel and more effective forms of drug delivery have indicated the mucosal approach as a promising non-invasive therapeutic pathway for systemic drug delivery. In addition to non-invasive and painless administration, mucosal delivery can provide several benefits, such as easy accessibility, rapid onset of action, elimination of the hepatic first-pass effect, high bioavailability, low cost, self-administration and good patient compliance, thus representing a promising and valid alternative to the parenteral route [1,2,3]. Moreover, mucosal delivery offers different absorptive surfaces, i.e., buccal, nasal, ocular, vaginal and rectal mucosa, thus providing great opportunities not only for systemic, but also for local administration of a variety of compounds, giving the possibility of targeting the specific tissue and obtain high “in situ” drug concentrations and reduced systemic side-effects in the local treatment of diseases [3,4]. Mucosal drug delivery can be also obtained by the oral administration of suitable mucoadhesive delivery systems, exploiting the mucosa lining of the gastrointestinal tract [5].

Parallel to the growing interest in mucosal delivery, mucoadhesive polymers became increasingly attractive as a platform for the development of mucosal delivery systems [6,7,8,9], and, among them, a dominant role is played by chitosan (CS) [10,11,12].

CS is a linear polysaccharide of natural origin, obtained by full or partial deacetylation of chitin, the second most abundant biopolymer in nature after cellulose, being the primary component of the exoskeleton of marine organisms (such as crabs, lobsters and shrimps), which represent its main commercial sources [13,14] CS is a copolymer composed of randomly distributed β-1,4-linked D-glucosamine and N-acetyl-D-glucosamine units (Figure 1).

CS is practically insoluble in water, but it becomes soluble in acidic solutions where its amino groups become protonated. Its cationic character, which renders it unique among biodegradable polymers, is considered responsible for many of the properties of CS, including its ability to strongly interact with different types of molecules, such as the anionic components of the mucosal surface [10,11,12,15], as well as to form polyelectrolyte complexes (PECs) with a variety of negatively charged polyanionic materials [16,17,18].

Since joining the pharmaceutical field, in the early 1990s, CS attracted a great and constantly increasing interest, and is presently considered as one of the most versatile natural polymers, being endowed with many favorable biological, biomedical, biopharmaceutical and technological features, along with an absence of toxicity [19,20,21]. On one hand CS exhibits several biological activities [22], including antifungal and antibacterial [23,24,25,26], anti-inflammatory [27,28], antitumor [29], immuno-enhancing [30], antioxidant [31], hypocholesterolemic [32,33], hemostatic [34] and wound healing [35,36] properties.

On the other hand, its wide availability, low cost, non-toxicity, biocompatibility, biodegradability, ease of chemical modifications, mucoadhesion and permeation enhancer power, joined to its capability to form films, hydrogels and micro- and nanoparticles, make CS an advantageous excipient not only for the development of a variety of delivery systems [37,38], but also in gene delivery [39], tissue engineering [40] and food technology [41]. Moreover, in order to improve its physicochemical properties and stability and further extend its applications, several derivatives of CS were investigated, obtained by both chemical modifications [12,42,43,44] or by polyelectrolyte complexes (PECs) formation [16,17,18,45,46].

Even though the use of CS was proposed for a very wide range of drug delivery systems, there is no doubt that the primary interest in CS-based systems is currently mainly centered on mucosal drug delivery, due to both the increasing importance of the mucosal route as an effective alternative method of administration, and the unique biological properties of CS, particularly suitable, as we will comment later, to favor and improve mucosal delivery.

Different reviews proved the actual importance and great potential of CS and its derivatives in mucosal delivery [10,11,12,46,47]. However, there is not a comprehensive review about the specific and distinct contributions which can be provided by CS in the variety of especially developed delivery systems intended for the different administration sites offered by the mucosal route. Therefore, this review aims to provide an updated overview of the peculiar applications of CS and its derivatives in the different kinds of formulations intended for the different sites of mucosal drug delivery, with particular emphasis on its uses in the development of innovative delivery systems.

Accordingly, the main properties making CS a base component in formulations for mucosal delivery will be firstly briefly commented on, and then its applications in buccal, nasal, ocular, rectal, vaginal and oral delivery will be illustrated, with a number of related examples showing the specific potential of this unique polymer in addressing the challenging aspects, and solving the inherent limitations presented by each of these different drug administration sites.

## 2. Properties of CS Making It a Good Candidate for Mucosal Drug Delivery

CS is a very promising biomaterial in mucosal drug delivery, not only by virtue of its several favorable biological features, the most important of which are commented on in more detail below, but also due to its inherent biological activities, such as, in particular, its wound healing, anti-inflammatory and antimicrobial properties [26,27,36].

### 2.1. Muco-Adhesiveness

Mucoadhesion is a complex phenomenon mainly involving interactions with mucin, a complex mixture of glycoproteins representing an essential component of the mucus, the protective barrier lining all mucosal surfaces of the human body.

Mucoadhesive ability, i.e., the capacity to adhere to mucous membranes, is an important requisite for the development of an effective mucosal drug delivery system, since it can allow an intimate contact and prolonged residence time of the dosage form at the targeted sites, and then a gradual release and absorption of the active ingredients. The main expected advantages of mucoadhesive drug delivery systems are: site-specific drug delivery; sustained/controlled release; improved drug bioavailability (by virtue of the extended contact time with the mucosal surface) and consequent reduction of drug dose and dose-related side effects.

Many studies reported on the potent mucoadhesive properties of CS and on its beneficial effects on drug absorption improvement, as shown in a recent review [48], even though the detailed mechanisms behind these properties as well as the role of the peculiar structural features of the polymer remain not completely understood. Both the intrinsic properties of the interacting species, CS and mucin, as well as the environmental conditions, are involved in the complex phenomenon [49]. CS–mucin interactions appeared to be mainly caused by electrostatic attraction forces between the cationic polymer and the negatively-charged glycoproteins of mucin, complemented by other forces, such as hydrogen bonding, as well as hydrophobic interactions [50,51]. CS mucoadhesion power seems to be directly related to its deacetylation degree (DD), which affects both the number of free amino groups and the overall polymer conformation and chain flexibility [11,51]. Extrinsic factors which can affect CS–mucin interactions are the polymer concentration and the ionic strength and pH of the medium [49]. In particular, CS concentration should be below its critical concentration, beyond which its chains tend to overlap, forming a complex network, and then they are no longer free to interact with the mucin glycoprotein chains [52]. On the other hand, high salt concentrations could have an adverse effect, since they could screen electrostatic forces on the macromolecules surface, thus nullifying the contribution of electrostatic interactions between CS and mucin [52]. Finally, the pH of the medium is a critical factor to be considered, since CS mucoadhesiveness, being strongly related to its cationic nature, may occur only at acidic pH (pH < 6) [53]. This could be a drawback, giving rise to possible limits in its uses, particularly in gene or protein/peptide delivery, as well as to problems of CS precipitation when its acidic solutions will encounter neutral to basic pH environments once they are topically or systemically administered, with consequent variations in the carrier system performance. To overcome these issues and further improve its mucoadhesive properties, taking advantage of the ease of its chemical modification, CS derivatization was widely investigated to develop several modified versions of the polymer, endowed with customized physicochemical features [12]. Trimethylated-CS [54] (Figure 2) and thiolated-CS [55,56] (Figure 3) are among the more extensively utilized and favored mucoadhesive CS derivatives.

### 2.2. Biocompatibility, Biodegradability and Pharmacokinetics

The term “biocompatibility” substantially denotes the property of a particular material renders it compatible with living tissue, that is, does not generate a toxic or immunological response when exposed to the body or bodily fluids. It also denotes that the material efficiently and positively interacts with the biological environment of the human body. In other words, to be considered biocompatible, a biomaterial should carry out its functions giving rise to the desired effect without evoking any adverse reaction. The lack of any local irritant or harmful effect of the drug delivery system on the applied mucosal surface is a key requirement for allowing its safe, effective and repeated use.

One of the main favorable features of CS is that it performs its beneficial functions without inducing any immune response or inflammation or any other significant adverse effect on the biological system. Biocompatibility and the absence of cytotoxicity of CS carriers are reported in several reviews [20,57,58].

The exceptional biocompatibility and non-toxicity of CS are mainly attributed to the analogy between its chemical structure to that of glycosaminoglycans, which are one of the principal components of the human extracellular matrix [11].

This same favorable similarity also contributes to CS’s biodegradability. In fact, due to its structural similarity to the physiological glycosaminoglycans, CS is easily degraded in vivo by the hydrolytic action of lysozyme, a non-specific enzyme largely present in the mucus, and of chitinases and N-acetyl-D-glucosaminidases, enzymes produced by the colon-residing bacteria. The susceptibility to enzymatic depolymerization is an exclusive characteristic of CS with respect to other polysaccharides. Its final degradation products are D-glucosamine, N-acetyl-glucosamine and N-acetyl-glucose, which are all nontoxic to the human body; moreover, its degradation intermediates do not give rise to problems of accumulation in the body and do not have immunogenic power [59]. This allows the safe administration and degradation of topically applied CS-based mucosal delivery systems.

CS biodegradation rate mainly depends on its polymer molecular weight (MW), deacetylation degree (DD) and the pattern of N-acetyl-glucosamine residues; moreover, it is also inversely related to the polymer crystallinity, which is maximum for fully deacetylated polymer and has the lowest values for intermediate DD values, around 60% [60]. In addition, chemical modifications of CS can clearly significantly affect its biodegradation rate.

Finally, as for the CS pharmacokinetic behavior, the data that emerged from inherent pharmacokinetic studies indicate that its intestinal absorption is affected by the polymer MW (increasing with MW decreasing), and that, following both oral or other routes of administration, CS is rapidly eliminated in the urine, without causing significant accumulation in the body, as reported in a recent review [61]. Therefore, pharmacokinetics’ data, together with its non-toxicity, confirm the safe use of CS as a pharmaceutical excipient.

### 2.3. Permeation Enhancer Ability

CS is considered one of the most effective available polymers endowed with permeation enhancer activity, being at the same time non-toxic, non-irritant, biocompatible and biodegradable.

The CS permeation enhancing properties were mainly attributed to the positive charges of the polymer, which allow its interaction with the cells membranes, resulting in a structural reorganization of the tight-junction proteins, with a consequent reversible opening of the junctions, which favors drug permeation without negatively affecting cell viability or provoking any membrane injury [62,63]. The CS enhancement effect is clearly powered by the prolonged residence time of the drug at the mucosal surface, provided by the polymer mucoadhesive effect.

The structural features of CS, such as, in particular, its deacetylation degree and molecular weight, can significantly affect its permeation enhancer ability. In fact, an increase in the CS permeation enhancing effect was observed with an increase in both its deacetylation degree and molecular mass [11,62].

The CS permeation enhancing effect can be further improved by its suitable chemical modification. In particular, thiolated-CS derivatives (see Figure 3) not only exhibited improved mucosal adhesion, by virtue of possible additional interactions with the mucosal surface, due to the formation of disulfides with cysteine residues of mucins [55,56], but also showed a higher permeation enhancer effect than unmodified CS. This last effect was attributed to the powered ability in tight junctions’ opening by the interactions of thiolated CS with the thiol groups of cysteine molecules, largely present on membrane receptors and enzymes [55].

### 2.4. In Situ Gelling Properties

The ability of CS to form hydrogels, by physical or chemical crosslinking, is well known, and it was widely exploited for the development of a variety of drug delivery devices able to provide a local and sustained drug release [38,64]. Chemical hydrogels were obtained, for example, by CS crosslinking via covalent bonds with other polymers, such as hyaluronic acid [65], polyvinyl alcohol [66] or by UV irradiation [67]. On the other hand, physical “reversible” hydrogels can be obtained by the formation of polyelectrolyte complexes (PECs) based on the spontaneous establishing of electrostatic interactions between the protonated ammonium groups of CS and a variety of oppositely charged polymers, particularly from a natural origin, such as gums, alginates, pectins, carrageenans, etc. (Figure 4) [46].

Physical hydrogels are generally preferred, since they allow the preservation of all of the favorable properties of CS, including biocompatibility. Moreover, PECs’ formation can also be useful to enhance CS stability and suitably tune the polymer hydrolytic degradation rate.

Among these different types of chemical or physical CS hydrogels, stimuli-responsive “in situ” gelling systems can be obtained, resulting in free-flowing liquid formulations able to undergo a rapid transition into a gel phase on the site of interest, triggered by factors such as pH or temperature changes at the physiological conditions [68].

In situ gelling mucoadhesive polymers, such as CS and its derivatives, can be very useful for improving the effectiveness of mucosal drug delivery. In fact, on the one hand they allow an easy application of the liquid formulation at the level of the different administration sites; on the other hand, their rapid in situ gelation at the site of interest results in a very strong viscosity increase, that improves the polymer mucoadhesive effect and prolongs the mucosal residence time, minimizing shortcomings such as fast clearance or drainage of the drug from the target mucosa [69,70].

## 3. Limitations of Chitosan

Despite its several favorable properties and the extensive range of potential applications it can offer, the actual pharmaceutical and biomedical uses of CS are still bound by some important issues.

The very low solubility of CS in aqueous solutions at a pH higher than 6.3–6.5 (corresponding to its pKa value) [11,49,53] is regarded as the major drawback for CS applicability in different fields of drug delivery, including in the case of mucosal drug delivery systems [10,12,19,21,42,54,59]. In fact, due to its basic nature, CS can exert its mucoadhesive ability, essential to prolong the residence time of the delivery system on the application/absorption site, only at an acidic pH (pH < 6), when it becomes soluble, due to the ionization of its amino groups [53].

Fortunately, both the limited mucoadhesive strength and the almost insolubility of CS at a near neutral pH can be easily overcome by the use of CS salts [10] or a variety of derivatives endowed with improved solubility and mucoadhesive properties, easily obtained thanks to the simplicity of CS chemical modification [12,42,54,55,56].

However, the advantages offered by the simplicity of the chemical modifications of the polymer is turned into the disadvantage of having had an exponential increase of new derivatives of remarkable variability, all needing an adequate chemico-physical and biopharmaceutical characterization before their proper and safe use [71]. Moreover, the cost of CS derivatives is often higher than that of the original polymer, making them less competitive compared to synthetic polymers with similar properties, available at lower cost and greater purity degree [21].

The problem of the purity also arises for the unmodified polymer. In fact, even though CS is a very abundant and easily accessible biomaterial from marine/food industry waste [14], it should be considered that high cleansing and purification processes are necessary before it can be used for biomedical or pharmaceutical purposes, for eliminating possible impurities of a various nature (proteins, pigments, pyrogenic agents, metals and other inorganics, etc.) that could compromise its safe application [19,21].

Another important issue relates to the very large variety of components of the CS family, differing in molecular weight, acetylation degree and acetylation pattern (i.e., the distribution of comonomers sequences), which should all be accurately defined, since they strongly influence its physicochemical properties and biological activities. In fact, if both the structural variability and the simplicity of chemical modification are a point of strength of CS, responsible for its high versatility, they are, however, reflected in a difficult characterization and an unclear understanding of how the combined effect of these complex variables affect the properties of the final products [71]. In summary, an adequate characterization should be considered essential and indispensable to exploit the properties of this important polymer to their best end. The lack of a suitable characterization is not only responsible for the conflicting experimental results often reported in literature [70,71], but also prevents the successful implementation of the “Safe-by-Design” strategy to the development of CS-based drug delivery systems [72].

## 4. Chitosan-Based Mucosal Drug Delivery Systems

### 4.1. Buccal Drug Delivery Systems

The delivery of drugs via the buccal mucosa, for local and/or systemic effect, has acquired an increasing relevance over the last decades, as proved by recent specific reviews [73,74]. The buccal mucosa can offer many advantages compared to the conventional oral administration route. In fact, while maintaining the favorable characteristics of non-invasiveness, painlessness, ease of accessibility and self-administration, it can provide the direct arrival of the drug into the systemic circulation, avoiding hepatic metabolism and possible drug degradation in the gastrointestinal tract, thus being particularly convenient for sensitive drugs such as peptides; moreover, its rich vascularization, reduced enzymatic activity and high permeability of the thin epithelium lining make it an attractive absorption site [3]. On the other hand, buccal drug administration can also be effectively exploited to obtain a local action, in the treatment of diseases such as oral mucositis, candidiasis, gingivitis, periodontitis, etc. In fact, depending on the desired effect, buccal mucosal delivery systems can be designed to give unidirectional release (into the oral cavity or across the oral mucosa), or multidirectional release (into both directions). The possibility of easy dosage form removal, to quickly interrupt the treatment in case of the appearance of adverse reactions, is a further advantage.

However, despite all these advantages, there are also some limitations linked to this administration route, such as limited absorption surface, mucus turnover time, low solubility of some drugs in the buccal environment and their possible degradation by enzymes present in the saliva [9]. In addition, the continuous saliva secretion as well as mouth movements and masticatory effects, during both food eating or drinking, can give rise to possible ingestion or unintentional removal of the buccal delivery systems, representing a further challenge to be faced in their development.

In order to overcome these shortcomings, buccal delivery systems should be able to remain at the application site for the desired time, unaffected by environmental conditions and salivary flow, to suitably control drug delivery and enhance drug permeation. The role of mucoadhesive polymers in the development of effective buccal delivery systems is therefore essential [75,76]. The beneficial properties of CS, such as in particular its bio-adhesion and permeation enhancing effect, joined to its biocompatibility, biodegradability, absence of toxicity or irritant effects, wound-healing and antimicrobial activity, as well as its formulation versatility, make it an optimal choice in the development of a variety of bucco-adhesive drug delivery systems [9,77].

The ease of CS chemical modifiability was widely exploited to further improve its effectiveness in oral mucosa delivery. CS derivatives, such as trimethylated and thiolated ones (see Figure 2 and Figure 3) are shown to be very effective in promoting the absorption through the buccal mucosa of hydrophilic macromolecules and peptides [62].

Novel sulfhydryl-linked CS conjugates (Figure 5) intended for buccal application, endowed with significantly enhanced stability and adhesive properties compared to the unmodified polymer, were recently obtained [78]. They were successfully used for the development of a buccal adhesive system for release into the oral cavity of pilocarpine in the treatment of xerostomia, allowing a better in situ adhesion and a more controlled release than the unmodified CS, by virtue of their lower degradation rate [79].

Mucoadhesive tablets were widely used as forms of buccal dosage, particularly to provide systemic drug release [80,81], even though they present potential disadvantages, such as patient discomfort due to unpleasant foreign body sensations and local irritation. Mucoadhesive tablets of CS, in mixture with other polymers such as Carbopol 940 [82] or gelatin [83], were prepared by direct compression for buccal delivery of carvedilol or propranolol hydrochloride, respectively; in both cases, the technological and functional properties of the tablets were found to be suitably tunable by varying the CS/polymer weight ratio.

Mucoadhesive films are one of the most innovative and interesting types of buccal dosage forms [84]. An ideal buccal film should be flexible, elastic and properly shaped and sized, to assure good comfort to the patient, and endowed with good bioadhesive power, to guarantee adherence to the oral mucosa. Several kinds of CS-based films designed for buccal drug delivery were realized, often using this biopolymer in blends with a variety of other polymers in order to overcome its drawbacks and improve its effectiveness, mainly in terms of mechanical, swelling and mucoadhesive properties, permeation enhancing effect, and controlled release behavior. For example, Timur et al. developed mono- and bi-layered mucoadhesive films using CS alone, or in mixture with hydroxypropyl methylcellulose (HPMC) as local delivery platform for the therapy of oral mucosal infections; the combined presence of both polymers allowed a strong increase in tensile strength and enhanced the antimicrobial activity of the model drug, cefuroxime [85]. The incorporation of CS in polyvinyl alcohol-based buccal films improved the formulation performance, enhancing the mucoadhesion and permeation properties [86]. A buccal film based on CS in mixture with polyvinylpyrrolidone (PVP) was proposed for the local delivery of tenoxicam in the treatment of chronic periodontitis; the final formulation, optimized by factorial design, exhibited satisfying mechanical and bioadhesive properties and allowed a continuous drug release at therapeutic concentrations for 6 h, at a dosage lower than the oral one [87]. Abouhussein et al. compared the effectiveness of a series of buccal films based on CS alone or blended with HPMC, methylcellulose, hydroxyethyl cellulose or polyvinyl alcohol (PVA) for local cetylpyridinium release as an antibacterial agent for oral diseases’ treatment; the best results were obtained for the films containing PVA, whose presence reduced the excessive swelling of CS, and improved both tensile strength and mucoadhesion [88]. A composite film based on CS in combination with a physically modified starch was developed for the buccal delivery of hydrophilic drugs, where the addition of CS increased tensile strength, reduced excessive swelling and enhanced permeation compared to the starch alone [89]. The positive effect of cyclodextrins in the performance improvement of CS films aimed for the buccal delivery of poorly soluble drugs was also proved [90,91].

Other strategies to improve the performance of CS buccal films are based on its ability to form polyelectrolyte complexes (PECs) with different negatively charged polymers. For example, Pilichera et al. realized a multilayer mucoadhesive buccal film by layer-by-layer deposition of casein sodium salt and CS on a polylactic acid substrate; the authors showed that a following CS double-crosslinking with glutaraldehyde and Na tripolyphosphate further improved the film’s performance, particularly in terms of mucoadhesion, better drug loading capacity and release control [92]. More recently a novel buccal mucosal delivery system based on a CS-sodium alginate (SA), PEC film and a water-repellent layer of ethyl cellulose to provide unidirectional drug release, was developed; a comparison with corresponding films made with CS or SA alone revealed the superior properties of the PEC film, in terms not only of mechanical properties, but also of drug release and ex vivo permeation through rabbit buccal mucosa [93]. Moreover, in vivo studies on rats showed that the developed CS-SA PEC film loaded with zolmitriptan and etodolac as model drugs presented a relative bioavailability of 246% and 142%, respectively, compared to the oral administration [93].

CS wafer formulations were investigated as an alternative to films. For example, a CS–sodium alginate PEC wafer for the buccal delivery of macromolecules was developed, using albumin as model drug [94]. However, CS wafers were less studied and appeared less attractive than films, probably due to the longer time and higher cost of their preparation, due to the lyophilization step, as well as to the limited advantages presented compared to the respective film formulations [85].

Even mucoadhesive patches, consisting of, typically, laminated non-dissolving matrices with a mucoadhesive layer, have obtained great attention as buccal delivery systems, particularly for systemic drug release [84]. A buccal patch for insulin delivery, based on CS as mucoadhesive matrix and “ionic liquids/deep eutectic solvent” (made from a choline–geranic acid mixture) as transport facilitator, was successfully developed [95]. The combined use of CS and PluronicF127 was successful in improving mucoadhesive power and in situ residence time of a metoprolol-loaded buccal patch, allowing a drug sustained release of up to 8 h [96]. An innovative multilayer buccal patch, obtained via a layer-by-layer process by electro-spraying of phenylalanine nanoparticles containing metronidazole into a mucoadhesive CS/polyvinyl alcohol/ibuprofen electro-spun patch was realized, which provided a controlled dual-release of the loaded drugs and showed good hemostatic activity and optimal antibacterial action [97].

The use of CS hydrogels for sustained drug delivery are widely exploited, including in the field of oral mucosal drug delivery, as shown in recent reviews [64,98]. CS hydrogels can also be suitably functionalized to further improve their performance. For example, hydrogels for buccal drug delivery, prepared with a catechol-functionalized CS, chemically crosslinked with genipin, showed that the presence of catechol improved the mucoadhesive power with respect to the simple genipin-crosslinked CS hydrogel, increasing the in vitro residence time from 1.5 h up to 6 h, and reducing the erosion rate; moreover, higher serum concentrations and a more prolonged release pattern of the loaded model drug lidocaine were obtained after application on the buccal mucosa of rabbits of the catechol–CS hydrogel, compared to the simple CS hydrogel [99].

Environment-sensitive CS hydrogels were also developed as drug delivery systems in the treatment of oral diseases. For example, glucose-sensitive CS-polyethylene oxide [100] or photo-crosslinked CS hydrogels [101] loaded with metronidazole, allowed the modulation of the release of the antimicrobial drug as a function of the glucose concentration, resulting in a promising treatment of diabetics’ chronic periodontitis. Even thermo- and pH-responsive carboxymethyl–hexanoyl–CS hydrogels were n developed, that proved to provide an efficient delivery of naringin, resulting in the effective treatment of periodontitis [102].

Mucoadhesive micro- and ever more nano-particles are emerging as an interesting approach for buccal drug delivery, since their high surface area enable them to attain an intimate contact with a large mucosal surface area; moreover, they can provide sustained drug release and adequate protection of sensitive drugs, thus enabling peptide and protein delivery also through the buccal mucosa, exploiting the advantages offered by such an administration route [103].

Further advantages can be obtained by developing composite systems where micro- or nanoparticles are embedded in buccal films or hydrogels, thus exploiting at the same time the benefits of both kinds of formulations [104]. For example, CS microparticles containing a bioactive peptide with antihypertensive properties were loaded in a CS mucoadhesive film, thus combining the CS film’s ability to promote penetration of large molecules through buccal mucosa with the protective effect of drug entrapment into CS microparticles [105]. A systematic study was performed to assess the potential of a nanocomposite CS film impregnated with PLGA nanoparticles containing the enriched flavonoid fraction of Cecropia glaziovii (EFF-Cg) as an effective buccal delivery system of this phyto-pharmaceutical product in the treatment of herpetic infections; the developed nanocomposite film showed adequate mechanical properties, and absence of cytotoxicity, representing a promising strategy for overcoming the problems of the low EFF-Cg bioavailability and allowing its effective buccal delivery [106]. Nanoparticles composed of thiolated-triethyl-CS and embedded into a mucoadhesive CS film were developed as a possible approach for insulin buccal delivery; ex vivo permeation experiments with rabbit buccal mucosa showed that nanoparticles made with this new CS derivative were more effective not only than the corresponding ones with the unmodified polymer, but also than those with triethyl-CS, allowing a marked increase in insulin permeability, which reached almost 100% after 480 min [107]. An experimental design study was performed to optimize the composition of insulin-loaded nanoparticles prepared by ionic gelation of thiolated-N-dimethylethyl–CS conjugate with Na tripolyphosphate; the optimized nanoparticles were then embedded in a mucoadhesive CS-gelatin bilayered film [108]. CS-alginate core-shell-corona-shaped nanoparticles containing dimethylfumarate loaded in an orodispersible film were developed, which showed a 1.6-fold increase in AUC_0−4_ and AUC_0−∞_ values with respect to the conventional oral film formulation, even at a very lower drug dose (2 mg/film vs. 30 mg/film), allowing a reduction of drug dosage and of the related adverse effects in the treatment of multiple sclerosis [109].

A different nanotechnological approach intended for buccal drug delivery was also attempted, based on mucoadhesive liquid crystal precursor systems (LCPSs), which are liquid formulations able to rapidly incorporate water from the saliva, becoming a viscous system, which can provide a controlled release rate in the buccal mucosa. Calixto et al. prepared various liquid crystalline systems containing ethoxylated/propoxylated cetyl alcohol as surfactant, oleic acid as the oil phase, and aqueous dispersions of CS, polyethyleneimine (PEI) and their combination as the aqueous phase; all the examined formulations behaved as LCPSs in the presence of artificial saliva and rheological, mucoadhesive and release studies of the incorporated peptide model drug indicated that the system containing the CS/PEI mixture was the most suitable for buccal application [110].

Sponges represent another interesting potential delivery system to give local or systemic drug release to mucosal surfaces, offering some advantages such as the ability to limit patient discomfort, due to their flexible and soft structure compared to buccal tablets, maintain their swollen structure for a longer time than hydrogels (and then provide longer in situ residence time), and allow a higher drug loading than thin films, owing to their highly porous nature. CS sponges of buspirone hydrochloride, prepared by a casting/freeze-drying technique, showed an ability to adhere to buccal mucosa for up to 8 h and provided a sustained release during this time [111]. More recently, CS-hydroxypropyl methylcellulose sponges, laminated with an external impermeable ethyl cellulose layer were developed for unidirectional buccal delivery of protamine-decorated tripterine phytosome; pharmacokinetic studies showed a strong increase in the bioavailability of tripterine from the new formulation with respect to its oral administration as a suspension, with a five-times increase in AUC, despite the 10-fold lower dosage [112].

An innovative formulation consisting of core-shell fibers able to self-assemble liposomes, designed for the systemic buccal delivery of carvedilol, was obtained by the coaxial electrospinning of a shell solution of the mucoadhesive component (carboxymethyl-CS-PVA or sodium carboxymethylcellulose-PVA) and a core layer solution (a mixture of drug and phospholipids); ex vivo permeation studies across a porcine cell culture model and buccal mucosa highlighted the role of the liposomes and the CS polymer in promoting the drug penetration [113].

### 4.2. Nasal Drug Delivery Systems

The nasal administration route is an attractive opportunity for local and systemic drug delivery as it provides a large, highly vascularized mucosa for drug absorption, avoids the first-pass metabolism, can provide a rapid onset of action, is easily accessible, non-invasive and painless and is suitable also for molecules labile in the GI medium [114]; moreover, it offers a direct gateway to the brain through the olfactory epithelium along the trigeminal and olfactory nerves [115,116].

However, despite its many advantages, the nasal route presents several factors that limit the drug absorption, such as the occurrence of mucus and epithelial barriers, the rapid muco-ciliary clearance, the possible degradation by mucosal enzymes and the poor permeability of hydrophilic molecules, including peptides or proteins [114]. Moreover, the limited volume which can be administered is an additional drawback, requiring highly potent drugs.

Several strategies were developed to improve the performance of nasal drug delivery systems, mostly aimed at overcoming the main barriers to an effective absorption, i.e., the short in situ residence time, due to the rapid muco-ciliary clearance, and the limited permeation through the epithelial membrane, by exploiting the use of both mucoadhesive excipients and adsorption enhancers. An ideal enhancer should induce a reversible increase in the transport of the drug across the nasal membranes, and remain adherent to the mucosa long enough to reach a maximum outcome, without causing irritation or toxic effects; on the contrary, many of these molecules (i.e., bile salts, surfactants and fatty acids) are often irritating or cause short- or long-term damage to the nasal mucosa [117].

Nanotechnological approaches represent an important tool for overcoming the problems of limited drugs’ absorption from the nasal cavity, and they gained increasing interest in recent years, allowing for the development of more efficient nasal drug delivery systems [11,118].

CS, due to its biocompatibility, non-toxicity and biodegradable properties, joined to its ability to adhere to the mucosa and to transiently open the tight junctions located between the epithelial cells, is considered to be an excellent candidate for the development of effective nasal delivery systems, being able to increase the in situ residence time of a formulation and at the same time to promote the transport of drug molecules through the biological membranes [119]. Moreover, the interest in using CS in nasal formulations is further increased by its ability to penetrate the Blood–Brain Barrier, which, coupled with the above-mentioned properties, makes it particularly advantageous as a brain-targeted drug carrier in nose-to-brain delivery systems [120,121].

An additional advantage offered by CS is its high versatility, which allows its use in many physical forms, including nanotechnological-based systems, such as micro- and nano-particles, hydrogels, nanoemulsions, etc. [122].

However, the very low solubility of CS at neutral or alkaline pH values, as well as its poor mechanical properties in wet conditions, have led to the synthesis of a variety of derivatives [12] or to its combined use with other polymers, often exploiting the formation of polyelectrolyte complexes (PECs) [46], all aimed at overcoming CS deficiencies and improving its physicochemical properties and stability.

In virtue of their biodegradability, biocompatibility, low toxicity, mucoadhesion, large surface area-to-volume ratio, good drug payload, controlled release and permeation enhancer properties, CS-based micro- and nanoparticles were profitably utilized for improving the effectiveness of nasal administration of several kinds of active agents, including antihypertensive [123], antiemetic [124,125], anti-asthmatic [126,127,128], antibiotic [129] or anti-inflammatory [130,131] drugs. The favorable properties of CS micro- and nanoparticulate systems can allow a reduction in the dose and frequency of drug administration and, consequently, its side effects. The use of suitable CS derivatives, as well as the optimization of the preparation techniques of such systems, can potentially overcome the limits of CS poor solubility, improve its mucoadhesiveness, and increase drug encapsulation efficacy [47].

Among the different classes of drugs investigated as possible candidates for intranasal administration, vaccines have emerged as one of the most promising ones. The nasal route resulted in a particularly attractive site for vaccine administration, due to the presence of nasopharynx-associated lymphoid tissue (NALT) which has a key function in antigen recognition and stimulation of immune response, inducing both mucosal and systemic immunity. Moreover, intranasal immunization requires lower doses of antigens than the oral one, because it allows lower enzymatic degradation.

Micro- and nanoparticles based on CS or CS derivatives, alone or in combination with other polymers (such as poly (lactic-co-glycolic acid or poly-γ-glutamic acid), or functionalized with mannose, were successfully developed as promising nasal delivery systems for different types of vaccines [132,133,134,135,136,137,138,139].

The particular effectiveness of the CS nanoparticulate systems as a vaccine delivery vehicle is mainly attributed to its positive charge, which allows its interaction with the negatively charged sialic acid present in the mucus, as well as to its ability to open the tight cell junctions, both leading to an enhanced permeation of the antigens; this is joined to its immunomodulation ability, which enhances the immunogenicity of the encapsulated vaccines, as well as to its adjuvant activity in the mucosal membrane [140,141,142,143]. Moreover, CS nanoparticles protect the antigens from enzymatic degradation and increase the vaccine uptake, due to their mucoadhesiveness and their consequent prolonged contact time with the bloodstream capillaries [144].

Another important application of CS nanoparticles is as a “nose-to-brain” carrier of drugs in the treatment of CNS diseases [115,116,145]. The nose-to-brain-transport, bypassing the blood–brain barrier (BBB) and allowing direct brain-targeting via the olfactory mucosa and trigeminal neural pathways, can enhance the efficiency of drug targeting and reduce systemic side effects. Due to its non-invasiveness, the nose-to-brain route is considered the most effective one to transport a drug to the brain, even though the low volume administrable (maximum around 200 μL) represents an important limitation of the amount of drug transportable into the brain by this route [145].

Incorporation of drugs in CS nanoparticles for brain delivery resulted in a high brain uptake in vivo, attributable to factors such as: high entrapment efficiency, due to interactions between drug and CS matrix; particle size, that affects their endocytosis rate through the endothelium of brain capillaries; high positive surface charge, that promoted interactions with the negatively charged cell membranes, increasing their residence time on the nasal mucosa, and with tight junctions of the mucosal epithelial cells, leading to their transient opening [120].

Rotigotine-loaded CS nanoparticles were developed aimed for nose-to-brain delivery in the treatment of Parkinson’s disease; it was found that the CS-nanoparticle formulation provided increased nasal residence time and a 2.67-fold enhancement in drug permeability with respect to the drug solution, allowing a reduction of the dosage regimen [146]. The use of CS-based nanoparticles for intranasal administration in the treatment of Alzheimer’s disease was recently reviewed [147].

The use of suitable CS-derivatives, such as the thiolated (see Figure 3) or carboxymethylated ones, or surface modifications of the CS nanoparticles with targeting ligands can further improve their efficiency in brain-targeted delivery [120,148].

For example, carboxymethyl–CS nanoparticles were developed for intranasal delivery of carbamazepine in the treatment of epilepsy; in vivo studies showed that the drug encapsulation in CS nanoparticles increased its bioavailability with respect to a simple drug solution (plasma AUC_0−∞_ and brain AUC_0−∞_ 2.6 and 8.1 times higher, respectively) and enhanced brain targeting (brainAUC_0−∞_/plasmaAUC_0−∞_ ratio 2.6 vs. 0.5), also providing a sustained release [149].

Some examples of carboxymethylated-CS derivatives are shown in Figure 6.

Galantamine-loaded thiolated-CS nanoparticles, intended for drug therapy in Alzheimer’s disease, were prepared and evaluated for intranasal delivery, in comparison with the oral or intranasal administration of a drug solution; the significant superior recovery of scopolamine-induced amnesia, obtained in a mice-model using the modified-CS nanoparticles, proved their better therapeutic effectiveness [150].

Decoration of the nanoparticle surface with CS, by covalent or non-covalent mechanisms, can be exploited to improve their biological and physicochemical properties and their potential for drug–brain delivery [151].

Coating with CS of simvastatin-loaded poly-ε-caprolactone nanoparticles conferred mucoadhesive and controlled release properties (affected by the molecular weight of the CS used) and increased by 9.9- or 6.6-fold (using, respectively, low-molecular weight or high-molecular weight CS) the drug amount that permeated after 4 h through a human nasal cell line compared to the control (drug suspension) [152].

Chatzitaki et al. developed CS-coated PLGA nanoparticles loaded with ropinirole hydrochloride, where the CS coating provided them highly mucoadhesive properties and markedly increased drug permeation through sheep nasal mucosa by 2.33-fold, in comparison with uncoated nanoparticles, resulting in a promising approach for Parkinson’s disease [153]. Nanostructured lipid carriers (NLC) loaded with berberine and then coated with CS showed increased permeation via the nasal mucosa and significantly higher drug levels in the brain in comparison to both intravenous or intranasal administration of a drug solution (AUC_brain_/AUC_blood_ 1.22 vs. 0.24 and 0.99, respectively) [154].

CS functionalization with suitable substances can further increase the nanoparticles’ efficiency as shuttles for direct nose-to-brain delivery of drugs. For example, CS coating of NLC increased their transport three-fold through an in vitro model of olfactory cell monolayer; coating with CS functionalized with cell penetrating peptides’ moieties allowed a further two-fold increase of the transported nanoparticles, compared to those coated with unmodified CS [155].

Another attractive approach for drug intranasal administration consists of the development of in situ thermosensitive CS-based hydrogels. These are easily administered as drops or a spray, and rapidly form a gel at the nasal mucosa temperature (35–37 °C), thus reducing the muco-ciliary clearance rate and, consequently, enabling a sustained release of the drug [119,156]. Moreover, the soft consistency of swollen hydrogels makes them non-irritant to the nasal mucosa. In order to obtain the sol–gel transition temperature, mucoadhesion and gel strength appropriate for nasal administration of the in-situ gel formulation, CS is often used in combination with other components, such as glycerol-phosphate, gelatin, poloxamer, etc.

For example, a thermosensitive in situ gel, consisting of a mixture of CS, poloxamers 407 and 188 and Carbopol 934P, was developed for the intranasal administration of the anti-Parkinson drug, rasagiline mesylate; in vivo studies in rabbits and rats evidenced a significant bioavailability enhancement (up to six-fold) compared to oral drug administration, and a significantly (*p* < 0.01) higher concentration in the brain tissue than the control nasal solution, demonstrating the actual effectiveness of the formulation [157].

The use of CS in combination with Carbopol 934, poloxamer 407, sodium carboxymethylcellulose and hydroxypropyl methylcellulose, allowed the development of an in situ gel for nose-to-brain delivery of rivastigmine tartrate, which showed a three-times increase of the drug permeation through the nasal membrane and a 3.4-times higher distribution to the brain compared to the simple drug solution [158]. The use of a mixture of CS, quaternized CS and gelatin allowed the obtaining of a thermosensitive hydrogel with low gelation time, suitable swelling ratio and extended release of insulin up to 24 h, thus demonstrating the potential application of the formulation to improve the absorption of hydrophilic macromolecular drugs through the nasal route [159].

The combined use of the in situ gel and suitable nanocarriers can allow the further improvement of the formulation effectiveness. For example, an in situ gel based on a mixture of CS and β-glycerol phosphate, containing nanostructured lipid carriers (NLCs) loaded with lorazepam was developed for direct nose-to-brain delivery of the drug; in vivo studies showed that the new “drug-in NLC-in gel formulation” was more effective than the corresponding “drug-in gel formulation” in reducing the pentylenetetrazol-induced seizures, proving its potential usefulness for brain targeting through the nasal route [160].

CS-based thermosensitive hydrogels were also successfully developed for the local treatment of nasal wounds [161] and allergic rhinitis [162].

Another interesting strategy exploited to improve the effectiveness of nasal drug delivery systems is based on the development of nanoemulsions (NEs), heterogeneous systems consisting of oily droplets finely dispersed in an external aqueous phase and stabilized by surfactant molecules. Intranasal administration of NEs represents a possible alternative to oral and intravenous administration of drugs, due to their solubilizing capacity and ability to encapsulate lipophilic drugs and protect them from enzymatic and chemical degradation. Moreover, thanks to their small size, large surface area and surfactant presence, NEs allow a fast and high drug permeation to the brain from the nasal epithelium by transcytosis/endocytosis mechanisms [163]. The globule size and Zeta potential of NEs play a determinant role for drug delivery to the brain: the smaller the dimensions, the greater the retention time in the nasal cavity [164]. Furthermore, NEs with positive values of Zeta potential showed high retention times as a result of their interaction with the negative charges present in the nasal mucosa [165].

CS was widely used as component of NEs, to extend their nasal residence time, due to its mucoadhesive power and ability to confer positive charges; moreover, its enhancer properties allow the achievement of greater mucosal absorption and a high influx of drugs from the nose to the brain [164]. Bahadur et al. developed buffered NEs of the antipsychotic drug ziprasidone hydrochloride and evaluated their potential for an effective nose to brain delivery: NEs containing 0.5% CS showed an about two-times higher diffusion coefficient and a significantly superior efficacy in both paw test and locomotor activity test, compared with the corresponding NEs without CS [166]. Curcumin NEs for intranasal delivery were developed, by adding or not adding a CS solution to the formulation; ex vivo diffusion studies through sheep nasal mucosa showed that the mucoadhesive NEs had a higher flux compared to the NEs without CS (359.9 ± 36.8 vs. 302.8 ± 29.8 μg/cm^2^.h), and this result was attributed to the penetration enhancing effect of CS, due to its ability to open the tight junctions of the nasal mucosa, as well as to its positive charge and mucoadhesiveness, which allowed a better interaction with mucus, enhancing the drug residence time [167].

The favorable effect of CS in the development of NEs aimed for nose-to-brain drug delivery was also proved in the treatment of different pathologies, including migraine [168], neurodegenerative diseases [169] and mental pathologies [170].

To further increase their residence time in the nasal cavity, NEs can be formulated in thermosensitive gels capable of reducing the muco-ciliary clearance rate, thereby ensuring a more prolonged drug release [171,172].

### 4.3. Ocular Drug Delivery Systems

The ophthalmic administration of drugs can occur through the topical route, by local injection and through systemic (oral or parenteral) routes, depending on the nature and location of the eye disease to be treated. Topical application is the most frequent route for ophthalmic drug administration as it provides a selective delivery, with a high therapeutic index, allows the avoidance of first pass metabolism, is not invasive and meets with good patient compliance [173]. However, its major disadvantage is the limited drug bioavailability, attributable to various precorneal factors (such as tear film and tear turnover, blink reflex, solution drainage, induced lacrimation and presence of metabolic enzymes in the lacrimal fluid or ocular tissue), and to anatomical barriers, mainly represented by the corneal one [173]. For these reasons, conventional ocular formulations generally show insufficient therapeutic efficacy and require the frequent administration of high drug concentrations, often provoking undesirable side effects.

The main strategies used to improve drug ocular bioavailability were focused on improving its solubility/stability, prolonging in situ residence time and enhancing corneal permeability by the development of suitable innovative formulations [174].

CS represents an excellent candidate for the development of ophthalmic formulations, due to its biocompatibility, absence of toxicity, biodegradability, mucoadhesiveness, antimicrobial activity, solubilizing and permeation enhancer properties, and its actual effectiveness is proved by several recent reviews [175,176,177,178]. Furthermore, CS was also shown to induce the migration of keratinocytes, thus improving corneal wound healing [175].

Several CS derivatives were also explored to overcome some limitations of the unmodified polymer, such as, in particular, its poor water solubility and lower mucoadhesion at neutral pH, where its amino groups are deprotonated, and their ability to interact with the negatively charged mucin is therefore limited. This problem is particularly important in the case of ocular formulations, since the pH value of the eye mucus is about 7.4–7.8. Different CS derivatives were obtained, endowed with good solubility and high mucoadhesion properties and also at a neutral pH [12,42].

Different CS-based delivery systems, including polymeric nanoparticles, CS-coated nanocarriers, CS-based in situ gels or “nanocarriers in CS-based in situ gels” were investigated for topical ocular drug administration.

CS-nanoparticles possess many advantages that make them suitable for ophthalmic drug delivery systems, such as a high specific surface area (and, consequently, enhanced mucoadhesion compared to the bulk polymer), protection of the entrapped active molecules from degradation, prolonged in situ drug persistence, ability to control drug release, and thereby improved therapeutic efficacy [179,180]. Moreover, their nanometric size permits the penetration of the drug through the ocular barriers without causing ocular irritation and patient discomfort [176].

CS nanoparticles were widely investigated as drug delivery carriers for the treatment of various eye diseases. For example, mucoadhesive CS-sodium tripolyphosphate nanoparticles developed for topical ocular delivery of acetazolamide in the treatment of glaucoma showed significantly higher hypotensive activity in vivo (*p* < 0.05) than the simple drug solution [181]. Mucoadhesive trimethyl-CS nanoparticles loaded with flurbiprofen as hydroxypropyl-β-cyclodextrin-complex were developed for the treatment of ocular bacterial conjunctivitis; they allowed the enhancement of both drug solubility and in situ residence time, thus improving the drug therapeutic efficacy and patient compliance [182]. Galactosylated-CS nanoparticles were more effective as carriers for the ocular delivery of timolol than the corresponding nanoparticles with unmodified CS, in terms of enhanced trans corneal penetration (P_app_ 9.45 × 10^−6^ vs. 6.92 × 10^−6^ cm s^−1^) and longer retention in the cornea; moreover, they showed better efficacy in intraocular pression lowering and a more prolonged in vivo action, compared to commercial eye drops (*p* ≤ 0.05) [183]. Yu et al. prepared glycol–CS–dexamethasone conjugates able to spontaneously self-assemble into nanoparticles and investigated their potential for local ophthalmic delivery; in vitro studies evidenced their great mucoadhesive power and extended release properties during 8 h, while in vivo studies on rabbits proved the absence of eye irritation phenomena, and longer permanence at the corneal surface compared to a simple solution [184].

The favorable properties of CS nanoparticles can be further improved by the combined use of this cationic polymer with suitable biocompatible, biodegradable, negatively charged polymers, such as lecithin [185], dextran [186], alginate [187] and hyaluronic acid [188], exploiting the formation of polyelectrolyte complexes (PECs). This approach can allow some advantages with respect to the use of CS alone, such as increased loading of the lipophilic drugs, enhanced mucoadhesive properties, better control of the encapsulated drug release rate and more limited burst effect.

On the other hand, CS coating of different types of nanocarriers can represent an interesting strategy to improve their properties, conferring a positive charge to their surface, making them mucoadhesive, and favoring cellular uptake, thereby obtaining more efficient ocular drug delivery systems. For example, CS-coated PLGA nanoparticles loaded with bevacizumab were developed, aimed to obtain a prolonged and targeted release to the posterior ocular tissue of this anti-VEGF (vascular endothelium growth factor) drug in the treatment of different ocular diseases; the significantly higher (*p* < 0.01) trans-scleral flux ((0.3204 ± 0.026 vs. 0.2510 ± 0.015 μg/cm^2^.h) and the extended release (>72 h vs. ≈24 h), compared to the simple drug solution, indicated the good potential of the formulation as an ocular drug carrier to the retina [189]. The CS coating of PLGA nanoparticles, developed as a carrier of triamcinolone acetonide in the treatment of retinal vasculopathies, allowed the controlled release properties of PLGA to join with the mucoadhesion ability of CS [190,191].

Selvaraj et al. developed a CS-coated nanostructured lipid carrier (NLC) formulation for topical ocular delivery of itraconazole; in vivo studies on rats showed that the CS-NLC formulation exhibited an anti-neovascularization effect, and good VEGF (vascular endothelium growth factor) targeting efficiency [192]. Li et al. prepared trimethyl-CS-coated lipid nanoparticles to enhance the ocular bioavailability of baicalein, to be used in the treatment of keratitis or glaucoma; the formulation showed a sustained in vitro release up to 48 h, absence of ocular irritation, more extended in vivo pre-corneal retention times and AUC values about three-times higher compared to the drug solution [193]. CS-coated polycaprolactone nanoparticles were developed for enhancing ocular delivery of dorzolamide; release studies showed an initial burst release for 2 h, followed by a sustained release up to 12 h, while ex vivo studies using excised goat cornea evidenced around a four-times increase in mucoadhesion strength (and thus, longer in situ residence time) and a two-times increase in drug permeation compared to the simple solution, both attributed to the presence of CS [194].

Tan et al. developed bioadhesive CS-loaded liposomes aimed to prolong the precorneal residence time and enhance the ocular permeation and bioavailability of timolol maleate; the formulation showed around a three-times increase in corneal permeation, longer in situ retention time and a higher reduction of the intraocular pression compared to the eye drops [195].

CS-coated liposomes were prepared as a potential carrier of triamcinolone acetonide in the topical treatment of posterior eye diseases and were compared with conventional liposomes [196,197]. The CS-coated liposomes exhibited greater entrapment efficiency, more extended retention time, probably by virtue of their highly positive charge, and more prolonged release than the corresponding conventional liposomal formulation; moreover, in vivo studies proved the actual penetration of the construct through the corneal mucosa barrier and its effective accumulation at the level of the vitreous body, and evidenced the presence of significant drug amounts retained in the eye after fifteen days of treatment [197]. Similarly, CS-coated liposomes showed more sustained in vitro release, better and more prolonged in vivo effects, and higher transduction efficiency into both corneal epithelial cells and retinal pigment epithelial cells than uncoated liposomes [196]. The results of both these studies confirmed the effectiveness of the CS coating in improving the performance of liposomal formulations, and highlighted the potential of these formulations as efficient ocular delivery systems to the posterior segment of the eye.

In situ gels represent an interesting strategy for ocular drug delivery, since, being liquid during instillation, they maintain the advantages typical of conventional eye drops, including ease of administration and dosing accuracy; however, they quickly change to the gel state after application on the ocular surface in response to environmental stimuli, thus allowing longer residence time of the drug to be obtained on the ocular surface, without the problems of blurred vision and poor patient compliance attributable to ointment [198,199]. Depending on the mechanisms causing the gelation phenomenon, there are three main types of in situ gelling systems, i.e., the pH-triggered, the temperature-triggered and the ion-triggered ones, as well as their possible combinations [198].

CS-based ophthalmic gels are very promising, due to the several favorable properties of this polymer, including, in particular, its mucoadhesive and penetration enhancing abilities [98]. CS is a pH-sensitive polymer, and, due to its cationic nature, it presents a sol–gel transition at a pH of around 6.5. The association of CS with other polymers can allow the improvement of the mechanical strength and rheological properties of gel formulations, and the better control of the drug release, that generally occurs by swelling, diffusion and/or erosion mechanisms [178]. The use of CS-based in situ gels, obtained by its combination with different stimuli-sensitive polymers, as a carrier for prolonged ocular drug delivery, was recently reviewed [175].

For example, a pH-triggered in situ gel was developed using a mixture of CS and Carbopol (a polyacrylic acid derivative) for timolol maleate; the association of the two mucoadhesive polymers improved the mechanical strength of the formulation and the ocular retention time, and allowed a controlled release of the drug over 24 h [200]. A CS–dextran sulfate association was proposed, able to jellify at pH 7.4, typical of lacrimal fluid, and providing a prolonged delivery of ciprofloxacin; the formulation showed a high entrapment efficiency and a controlled release over 21 h [186].

A thermosensitive gel based on the CS–gelatin-β-glycerophosphate association, and containing ferulic acid, was evaluated for corneal wound healing using an alkali-burn rabbit corneal model; a significant reduction of the injured zone surface was found after 3h from the application [201]. Another thermosensitive CS-gelatin-based hydrogel, loaded with levofloxacin as a topical eye drop formulation, was developed, aimed at providing a sustained release of the drug with a single application, effective at warding off common pathogens responsible for post-operative endophthalmitis; in vitro studies proved the biocompatibility (assessed by a viability test on corneal epithelial cells), sustained release of up to 7 days and long-term antibacterial activity of the formulation [202].

Combinations of CS with the thermoresponsive polymer poloxamer 407 were investigated to develop a thermosensitive gel containing neomycin sulfate and betamethasone sodium phosphate, to be used in the treatment of conjunctivitis; the optimized formulation showed proper gelling temperature, absence of ocular irritation and was suitable for assuring a prolonged release [203]. Kashikar et al. described a pH- and temperature- activated in situ gel as an ofloxacin ocular delivery system, based on a mixture of poloxamers 407 and 188 with CS and gellan gum; the in situ gel-forming ability of the formulation reduced precorneal drainage, increased the residence time in the eye, and enabled a 1.5 times C_max_ increase compared to an eye drop solution [204].

Gupta et al. developed pH- and ion-sensitive in situ gel formulations, based on the association of CS with gellan gum [205] or with sodium alginate [206] for the ocular delivery of timolol maleate or levofloxacin, respectively; CS turned into gel at the lacrimal pH (around 7.4), while the gellan gum and alginate jellified upon contact with the divalent cations present in the lacrimal fluid: this mechanism made the formulations able to resist the ocular drug drainage, and provide higher trans corneal permeability and longer precorneal retention times than conventional eye drops.

Exploiting a similar gelling mechanism, Imam et al. prepared an ocular in situ gel of besifloxacin using a combination of CS, gellan gum and polyvinyl alcohol, where this last polymer was added to improve the mechanical strength of the system [207].

Innovative in situ gel formulations were obtained containing a drug loaded into suitable nanocarriers, so as to combine and simultaneously exploit the advantages of both kinds of drug delivery systems. In fact, nanocarriers can favor the enhancement of precorneal residence time and corneal drug absorption, due to their nanoscale size, but, due to the low viscosity of colloidal dispersions, they can be quickly removed by drainage, tear turnover and blinking effects.

For example, CS nanoparticles, prepared by ionic gelation using sodium tripolyphosphate, were loaded with the antibacterial drug levofloxacin and then added to a sol–gel system based on a mixture of alginate and hydroxypropyl methylcellulose to enhance the corneal residence time after topical application; the obtained formulation was non-irritant and showed reduced lacrimal drainage and corneal clearance, better antibacterial activity (*p* < 0.05), longer in situ retention (*p* < 0.05) and higher cornea penetration (91.73 vs. 47.65 μm depth) than the drug solution, due to the combined effect of the high viscosity of the gel and the bioadhesive and permeation enhancer properties of CS [208].

Fabiano et al. prepared a thermosensitive ophthalmic gel based on a CS-β-glycerophosphate mixture, containing 5-Fluorouracil either as such, or loaded into CS-hyaluronate nanoparticles, and also evaluated the effect of partially replacing the unmodified CS with quaternized or thiolated derivatives; the best formulations (containing 20% quaternized CS) after instillation in rabbits gave a 3.5 times increase of 0–8h AUC compared to the simple eye drops, but the nanoparticle containing gel showed a prolonged zero-order release during 7 h, in contrast to the rapid peak followed by a slow release given by the gel containing the free drug [209]. In a following paper, the authors studied the impact of using different CS derivatives for preparing the nanoparticles to load into the thermosensitive gel: the best results were obtained with quaternary ammonium-CS conjugate and sulfobutyl-CS, both allowing a zero-order release up to 10 h, with around a 1.5-fold AUC increase compared to the nanoparticles based on unmodified CS [210]. This finding was attributed, in the case of the quaternized derivative, to the increased contact of the positively charged nanoparticles with the negatively charged ocular surface, and in the other case, to a slowed down release of the drug from the negatively charged nanoparticles, both resulting in improved ocular bioavailability [210].

Ahmed et al. incorporated polylactide-co-glycolide nanoparticles loaded with ketoconazole in different in situ gel formulations based on CS, either alone or in mixture with sodium alginate, poloxamer 407 or Carbopol 940, and always in the presence of 0.5% hydroxypropyl methylcellulose; the best results in terms of in vitro release, antifungal activity and trans corneal permeation were obtained with the CS-alginate combination [211].

A CS-gelatin based thermoresponsive gel loaded with latanoprost (as an anti-hypertensive agent) and curcumin-PLGA nanoparticles (as anti-oxidant and anti-inflammatory agent) was proposed as a dual-drug delivery system in the treatment of glaucoma; the formulation allowed a sustained release of both of the drugs for 7 days and effectively reduced the oxidative stress-mediated damage in the human trabecular meshwork cells [212].

Ibuprofen-loaded NLCs (nanostructured lipid carriers) were prepared and incorporated in a dual stimuli-responsive gel, based on the combination of the pH sensitive CS and the thermosensitive poloxamer 407, with the purpose of enhancing the ocular drug bioavailability and therapeutic efficacy; the polymers allowed in situ jellification and mucoadhesion on the ocular surface of the formulation, thus prolonging in situ residence time and enhancing drug release [213]. With the same purpose, dexamethasone-loaded NLCs were incorporated in an in situ ophthalmic gel, based on a mixture of hydroxypropyl-trimethylammonium chloride-CS with β-glycerophosphate [214].

Nanovesicles based on a surfactant mixture (Span 60/sodium deoxycholate) and loaded with acetazolamide were embedded into a CS-sodium tripolyphosphate gel to obtain an efficient ocular delivery of the drug; the obtained formulation showed a controlled release, good mucoadhesion time, minimal irritating effects and a more prolonged effect in the lowering of the intraocular pressure, compared to the commercial drug tablets (24 vs. 5 h, respectively) [215].

### 4.4. Rectal Drug Delivery Systems

Rectal administration represents a useful route of administration for the management of both local or systemic effects and an efficacious alternative to other delivery routes, especially the oral one, presenting some advantages over it. In fact, rectal administration can enhance the systemic bioavailability of many drugs, due to the possibility of avoiding the first-pass metabolism or degradation in the gastrointestinal tract or, again, to overcome gastrointestinal absorption difficulties. At the same time, higher drug levels can be achieved at the level of colorectal fluids and tissues, thus resulting in it being also suitable for the management of local diseases [216]. The rectal route is of particular application for the treatment of Inflammatory Bowel Diseases (such as ulcerative colitis, Crohn’s disease), anti-colorectal cancer drug delivery and delivery of biopharmaceuticals.

However, the poor acceptance of this route by patients, together with discomfort after administration, erratic and/or limited drug absorption and a limited retention time at the mucosal site represent the major drawbacks of the conventional dosage forms to be overcome for an efficient rectal drug delivery. The development of innovative formulations, also based on nanotechnological approaches, could allow the better exploitation of the advantages of the rectal administration route [217].

In situ gelling systems, liquid at room temperature and able to jellify at body temperature, and suitable mucoadhesive formulations, able to prolong the retention time at the local site and control the drug release, can represent promising approaches to improve drug therapeutic efficacy in rectal delivery. CS and its salts and derivatives can be usefully employed in this field, by exploiting their mucoadhesive properties and absorption enhancer effect, due to the tight junction opening. Moreover, the intrinsic wound-healing and anti-inflammatory activities of this polysaccharide make it even more suitable for the treatment of colorectal diseases.

Because one of the most important challenges for effective rectal drug delivery is the limited residence time at the mucosal tissue surface, mucoadhesive formulations such as hydrogels or nanogels containing CS were proposed to overcome this drawback. Xu et al. developed a mucoadhesive hydrogel, based on the catechol-modified-CS, crosslinked by genipin, to improve the efficacy of sulfasalazine rectal administration for the treatment of ulcerative colitis. Such an approach resulted in being more effective and safer than the sulfasalazine oral treatment, by avoiding the drug degradation to a toxic by-product in the upper gastrointestinal tract [218].

A combined approach exploiting both the beneficial properties of hydrogels and microparticulate systems was investigated by El-Leithy et al.: a mucoadhesive hydrogel formulation containing diclofenac-loaded CS microspheres after rectal administration to rats was able to remain attached to the rectal mucosa surface for up to 8 h, allowing a controlled drug release, and also a reduced irritation of the rectal mucosal tissue compared to both the free-drug-loaded hydrogel and the drug-loaded CS microspheres [219].

To develop effective mucoadhesive delivery systems for the local treatment of colorectal cancer, two types of doxorubicin-loaded CS-carboxymethylated/CS nanogels were prepared by electrostatic interactions between the polymers, followed by crosslinking with tripolyphosphate or calcium chloride, obtaining, respectively, systems with a negative or positive Zeta potential; positively charged systems showed improved mucoadhesion, and limited permeability, thus prolonging the in situ residence time and enhancing local drug concentration [220].

The ability of CS to establish electrostatic interactions was exploited to develop a 5-fluorouracil-loaded CS/polyacrylic acid/Fe_3_O_4_ magnetic nanocomposite hydrogel as a potential anticancer rectal delivery system; in vitro tests, simulating colon and rectal conditions, showed that the system enhanced drug stability and allowed its controlled release [221].

Among rectal formulations, mucoadhesive thermosensitive in situ gelling systems represent promising delivery systems, due to their easy administration, without damaging mucosal tissues, and their bioadhesive properties, which can provide suitable retention times and the possibility of controlling the drug release [222]. Thermo-reversible liquid suppositories based on poloxamer 407, containing CS microspheres loaded with candesartan cilixitil were successfully developed by Patil et al., enabling a good mucoadhesion in the rectum and a prolonged drug release of up to 12 h [223]. He et al. prepared thermo-reversible insulin liquid suppositories by using poloxamer as thermosensitive polymer and N-trimethyl-CS chloride (TMC) at two different degrees of quaternization (40% and 60%) as the mucoadhesive and absorption enhancer agent; the addition of both N-TMC types increased the gel strength and bioadhesiveness and, after administration to diabetic rats, gave rise to a better enhancer insulin absorption (with lower glucose levels) than sodium salicylate, at a 10-fold lower concentration (1% vs. 10%), without causing any irritation on the rectal tissues (as proved by histological studies) [224].

The effect of the TMC quaternization degree (L-TMC, 12% or H-TMC, 61%) on the in vivo absorption of insulin, after rectal administration in rats, compared to CS hydrochloride, was investigated by Du Plessis et al., demonstrating the important role of the CS quaternization degree on its absorption enhancer effect; in fact, while all the three polymers increased rectal absorption of insulin at pH 4.4, only H-TMC was effective also at pH 7.4, due to the insolubility of the CS salt in neutral environments, and the still too low charge of L-TMC at this pH value for promoting any significant interaction [225].

In addition to the mucoadhesive and absorption enhancer properties, the anti-inflammatory effects of CS were also well documented [27,28]. The exploitation of both the bioadhesive feature of the polymer and its own anti-inflammatory effect led to a successful formulation based on CS microspheres encapsulating mesalazine for the treatment of inflammatory bowel diseases; this approach allowed not only the drug release to be prolonged up to 48 h but also to increase its anti-inflammatory effect on the colon, following rectal administration, allowing a two-fold reduction of drug concentration with respect to a suppository commercial formulation [226].

In addition, Jhundoo et al. recently investigated the combined effect of CS with 5-aminosalicylic acid (5-ASA) in the treatment of Inflammatory Bowel Disease, evaluating the potential effect of the molecular weight and deacetylation degree of the polymer; the results obtained after rectal administration to colitis mice evidenced that the CS-5-ASA combinations, almost independently from the CS deacetylation degree and molecular weight, allowed a significant (*p* < 0.05) improvement of the anti-inflammatory activity, compared to CS or 5-ASA alone [227].

### 4.5. Vaginal Drug Delivery Systems

The vaginal administration of drugs can represent a valid alternative to the conventional oral route, particularly for the topical treatment of urogenital infections (including bacterial vaginosis, yeast vaginitis and urinary tract infections and other vaginal malfunctions), or for microbicide delivery against sexual HIV transmission [228,229], but also for systemic treatments, mostly for contraception or hormonal therapy [230]. The vaginal environment presents a relatively large surface area, high blood supply, good permeability, balanced pH and relatively low enzymatic activity, thus offering the possibility of achieving high localized drug concentrations, requiring a lower dose and thus reducing the side effects compared to the oral route. Moreover, in the case of drugs’ systemic absorption, the gastrointestinal side effects and first-pass metabolism can be avoided.

However, the self-cleansing action due to the mucus secretion, the humid environment of the vaginal lumen and the peristaltic activity of the vaginal wall can reduce the retention time of conventional vaginal formulations in the site of action, thus requiring frequent applications to assure the drug’s therapeutic efficacy. Moreover, the mucus layer, epithelia modifications and possible degradation by the enzymes present in vaginal fluid, represent further barriers to an effective vaginal drug delivery.

Mucoadhesive delivery systems, able to adhere to the vaginal mucosa by chemical and/or physical interactions with the mucus components, were proposed as a successful strategy to overcome the aforementioned drawbacks, by increasing the residence time and assuring a sustained drug release [231]. Various types of natural or synthetic polymers were used in the development of vaginal drug delivery systems, whose main role is to provide a prolonged residence of the system at the application site and a controlled drug release [232,233,234]. Among these polymers, CS appears particularly interesting, due to its favorable biopharmaceutical properties, including good tolerability, biocompatibility, biodegradability, non-toxicity and strong bioadhesive power; this latter is related to its high ability to interact with the negatively charged mucus components, owing to its cationic nature, and to strongly adhere to the mucosa, thus providing a controlled and sustained drug release. In addition to these properties, its antimicrobial and anti-inflammatory activities make it an optimal candidate for the development of vaginal drug delivery systems [231,234].

Moreover, by virtue of the ease of CS’s chemical modification, a variety of CS-derivatives were obtained, aimed to improve its physico-chemical properties, such as, in particular, its mucoadhesive strength and water solubility, as well as its stability in a wider pH range [12,54,55]. In particular, thiolated-CS showed clearly enhanced mucoadhesive, cohesive and permeation enhancer capabilities [55,56].

In the past few years, different drug delivery systems based on CS and its salts and derivatives were developed as suitable dosage forms for vaginal delivery, ranging from bioadhesive tablets to more innovative formulations, including films, hydrogels or various engineered systems exploiting nanotechnologies [235,236].

Mucoadhesive matrix tablets, where the drug is homogeneously distributed in a hydrophilic matrix, able to create an in situ gelling effect due to the interaction with the vaginal fluid are among the most common vaginal formulations, due to their ease of preparation, low cost and high stability. The combined use of CS, as key polymer, in physical blends with other mucoadhesive polymers, such as hydroxypropyl methylcellulose, sodium carboxy methyl celluose or guar gum, was exploited as an effective strategy to obtain bioadhesive vaginal tablets with prolonged release of the antifungal drug fluconazole [237]. Mucoadhesive vaginal tablets based on CS, alone and in combination with pectin and locust bean gum, were developed for the sustained release of the anti-HIV agent tenofovir; the best results were obtained with the CS–pectin association, where the formed polyelectrolyte complex (PEC) allowed a more robust gelled system to be obtained, able to remain attached to the vaginal mucosa and to provide a controlled and reproducible drug release up to 96 h [238]. CS-based mucoadhesive tablets loaded with Chelidonii herba lyophilized extracts showed high mucoadhesive properties and prolonged drug release, resulting in being suitable for vaginitis treatments [239]. CS, mixed with Carbopol 934 and sodium carboxymethylcellulose, was employed to realize the mucoadhesive slow-release layer of a double-layered vaginal tablet; while the fast-release effervescent layer was a lactose-maize starch granulated mixture, added with sodium bicarbonate, a biphasic release of the loaded Lactobacillus spp bacteria, used against vulvovaginal infections, was obtained, with a very rapid dissolution of the effervescent layer, followed by a prolonged release from the bioadhesive matrix layer for up to 24 h [240].

An entirely S-protected CS was synthesized, which showed high stability against oxidation and an increase in mucus viscosity (which is directly related to the mucoadhesive power) up to six-fold higher than unmodified CS, allowing the development of metronidazole vaginal tablets with longer mucosal residence time and prolonged release [241].

Polymeric films represent safe and effective alternative formulations for vaginal administration, assuring a prolonged in situ residence time and thus avoiding the need of frequent administrations. To achieve this goal, a suitable mucoadhesiveness is required. The use of CS in combination with other polymers was investigated in several studies for the development of vaginal films, in order to evaluate a possible positive effect in improving its mucoadhesive properties. For example, Abilova et. al. compared the performance of CS, alone or in association with poly(2-ethyl-2-oxazoline) (POZ), to prepare ciprofloxacin-loaded vaginal films; the presence of POZ allowed the drug release rate to increase, but decreased the film’s mucoadhesive properties [242]. On the other hand, Calvo et al. successfully combined CS and hydroxypropyl methylcellulose to develop tioconazole films for the treatment of vaginal candidiasis, endowed with excellent component compatibility and presenting antimicrobial activity without showing cytotoxic effects [243,244]. CS, in combination with xanthan gum, karaya gum or pectin, was employed to develop multi-layered vaginal films for release of the antiviral tenofovir, exploiting the CS ability to interact with anionic polymers to form polyelectrolyte complexes (PECs) able to assure a controlled drug release [245]. Eudragit^®^ L100/CS composite bilayer vaginal films were also developed for a pH-responsive release of tenofovir, by combining the CS mucoadhesive properties with the Eudragit^®^ L100 ability to improve the film’s mechanical properties and to control the drug release into the vaginal medium [246].

The association of CS with alginate was successfully exploited for developing a mucoadhesive membrane capable of encapsulating metronidazole and resulting in amembrane suitable for vaginal administration, due to its stability in a simulated vaginal fluid and ability to provide a prolonged drug release [247].

Hydrogels are soft three-dimensional polymeric networks entrapping a large amount of water within their structure. They can offer a great potential for vaginal application, by favoring the in situ residence time and controlling the drug release [248]. CS can form a variety of hydrogels by physical or chemical crosslinking [64,249], and this ability was also widely explored in the development of effective vaginal delivery systems. For example, a CS-based hydrogel containing a Mitracarpus frigidus extract was proposed as a possible therapeutic alternative for the treatment of *Candida albicans* vulvovaginal infections, being able to reduce the fungal load and the inflammatory processes with a great preservation of the morphological characteristics of vaginal mucosa [250]. A modified CS, obtained by crosslinking with BDDE (1,4-butanediol diglycidyl ether) and thioglycolic acid grafting, showed a two-fold stronger mucoadhesive power than the unmodified CS, and was successfully used to form a hydrogel for genistein delivery in vaginal atrophy treatment [251].

Thermosensitive hydrogels characterized by low viscosity at room temperature and gel transition at body temperature are particularly suitable for providing a quick spreading and effective coating of the vaginal mucosa. A promising formulation for vaginal candidiasis was developed by Abd Ellah et al., consisting of CS-gellan gum gel-flakes loaded with ketoconazole and then embedded in an in situ forming gel of poloxamer 407; the system was able to offer an efficient coating of the vagina, due to the free-flowing properties during application, the entrapment of flakes within the vaginal epithelia and prolonged drug release [252]. Mucoadhesive thermosensitive hydrogels for the topical treatment of Trichomonas vaginalis infections, composed of CS/poloxamer 407 and loaded with metronidazole [253], or of CS/poloxamers (407/188) and loaded with secnidazole [254] were obtained; in both cases, the hydrogel formulations showed good mucoadhesive strength, sustained release and a decrease in drug permeability through excised pig vaginal mucosa with respect to the control (a simple drug/CS or drug solution), thus enabling a reduction in systemic absorption and related side-effects. A CS-glycerophosphate thermosensitive gel was successfully developed, suitable for progesterone vaginal administration [255].

The development of mucoadhesive sponges able to be retained in the vaginal cavity and allow a sustained drug release were also considered. Sponges are dispersions of a gas (air) in a very porous solid matrix, which should maintain their swollen hydrated structure for longer than semi-solid hydrogels, thus providing more extended in situ residence. For example, buconazole nitrate-loaded vaginal sponges were prepared for the treatment of vaginal candidiasis, by lyophilization of the CS solution in mixture with gelatin, or hydroxypropyl methylcellulose (HPMC), using calcium chloride as the crosslinking agent and different solubilizers; the most effective sponge in terms of drug entrapment efficiency, mucoadhesion, retention time and drug release rate was composed of the CS/HPMC 1:1 [256].

The use of mucoadhesive multiparticulate systems can offer several advantages for vaginal administration compared to single dosage forms, owing to the possibility of covering a wider mucosa area, thereby allowing a prolonged in situ residence and a controlled release over time. CS-coated calcium alginate microspheres were successfully developed for cefixime [257] or metronidazole [258] vaginal administration, with excellent mucoadhesion, extended-release properties and good antibacterial activity, as proved by microbiological studies.

The higher potential of CS-based microcapsules (prepared by electrospraying) than conventional tablets as a delivery system for phytomedicines with antifungal and antioxidant activity for vulvovaginal candidiasis was evidenced [259].

Thiolated-CS-coated multilayer microparticles were effective for the vaginal delivery of topical anti-HIV microbicide, showing enhanced drug loading and higher mucoadhesion (20–50-fold) than uncoated microparticles [260]. CS glutamate seemed a good candidate to prepare spray-dried microbeads as a possible microbicide delivery platform for vaginal delivery of water-soluble drugs [261].

The combined approach exploiting nanotechnology and mucoadhesive polymers was extensively investigated in the field of mucosal delivery. Nanocarriers can offer several advantages, including improved drug stability against chemical/enzymatic degradation, enhanced bioavailability and a longer drug effect. Mucoadhesive nanoparticles made of CS or CS-derivatives [235,262,263,264,265], as well as CS-coated nanoparticles [266,267,268] were developed by several authors for a more effective treatment of vulvovaginal fungal infections, by simultaneously exploiting the benefits of both CS and nanostructured delivery systems.

An effective strategy for intravaginal peptide delivery was developed, consisting of mucoadhesive CS ascorbate nanoparticles loaded with insulin as model molecule, incorporated in suitable hydrophilic sponge-like cylinders to facilitate their application [269]. Acyclovir-loaded poloxamer-modified CS nanoparticles developed as a vaginal delivery system were more effective than the corresponding unmodified CS nanoparticles, showing greater drug entrapment efficiency, more sustained drug release and higher cellular uptake [270]. Arumugan et al. compared the performance of CS nanoparticles or marine sponge spicules (made of 99.9% of SiO_2_) as carriers for Callophicin-A against vaginal candidiasis, and proved the superior performance of CS nanoparticles in terms of both higher drug entrapment efficiency (65% vs. 38%) and superior antifungal activity [271].

Nanofibers represent another interesting nanotechnological approach, which exhibited high drug-loading efficacy, possibility of sustained release, together with ease of production and good cost-effectiveness [272]. For example, mucoadhesive, biocompatible and biodegradable CS/polyethylene oxide nanofibers, prepared by the electrospinning of the aqueous solutions of the polymers, and incorporating metronidazole, were proposed for a powerful treatment of local vaginal infections; the developed nanofibers showed high bioadhesion and long retention at the application site [273]. Meng et al. prepared, by a coaxial electrospinning technique, tenofovir-loaded nanofibers consisting of a thiolated-CS-polyethylene oxide internal core and a polylactic acid external shell; the core/shell nanofibers were 40–60-fold more bioadhesive than the corresponding ones without CS and allowed a 10-fold improvement in the drug loading percent compared to a nanoparticle formulation [274].

A recent strategy which attracted increasing interest consisted of combining the advantages of colloidal vectors and hydrogels, creating the so-called “nanogels”, where the drug-loaded nanocarriers are incorporated within the three-dimensional structure of the hydrogel. Nanocomposite hydrogels present different properties than their individual components, and generally form stronger and more elastic gels, beneficial to prolong the in situ residence and reduce drug leakage when applied to the mucosal surface [275]. This strategy was also successfully applied to improve the performance of CS hydrogels, as shown in a recent review [276]. Micro- or nanoparticles (loaded with suitable drugs) embedded in CS hydrogels can provide a further barrier to the drugs’ diffusion, delaying their release and limiting the burst effect.

For example, Yang et al. elaborated a thermosensitive hydrogel consisting of CS-glycerophosphate-sodium alginate where microparticles of CS/BSP (Bletilla striata polysaccharide), loaded with the anti-HIV drug tenofovir, were incorporated before the gelation process; the combined system drug-in microparticle-in CS gel showed not only a reduced burst effect but also superior mucoadhesion strength, 2.3 and 1.7 times higher than the drug-loaded microspheres or hydrogel, respectively [277]. A CS-carboxymethyl-CS nanogel loaded with rifaximin showed a pH-responsive swelling at acidic pH and enhanced bioadhesion, thus assuring a sustained release profile, together with an improved antioxidant activity and hemocompatibility [278]. Imiquimod-loaded poly(ε-caprolactone)-nanocapsules, coated or uncoated with CS, were inserted into a hydroxyethyl cellulose or a CS hydrogel; a comparison of the obtained nanocomposite hydrogels in terms of mucoadhesion, retention ability and permeation showed the superior performance of the drug-in nanocapsules-in CS hydrogel in the treatment of human papillomavirus infections [279].

Auranofin-loaded poly (lactic-co-glycolic) acid nanoparticles were embedded in a CS-β-glycerophosphate thermosensitive hydrogel for the effective topical therapy of vaginal infections; the composite system, due to the mucoadhesive properties of CS, showed a markedly enhanced local vaginal retention with respect to the free nanoparticles; its intravaginal administration in mice provided a nanoparticles’ retention longer than 6 h, and allowed it to reach markedly higher local drug levels compared to oral treatment with a much higher dose, with reduced plasma and liver levels, and then reduced systemic toxicity [280]. A biodegradable mucoadhesive nano-formulation was developed for vaginal candidiasis treatment with an antifungal essential oil, which was incorporated in a nanoemulsion and then dispersed in a CS hydrogel; the obtained formulation exhibited a lower irritant effect, superior mucoadhesive properties and higher antifungal activity than the free nanoemulsion, showing MIC values up to 64-fold lower [281].

Another type of combined strategy is to incorporate drug-loaded CS nanoparticles into suitable nanofibers, which should improve the bioadhesion and in situ residence time of the system. For example, Tugcu-Demiröz et al. prepared benzydamine-loaded CS nanoparticles embedded into polyvinylpyrrolidone (PVP) nanofibers or hydroxypropyl methylcellulose gel, and evaluated their potential for drug delivery against vaginitis; the developed nanoparticle-nanofiber or nanoparticle-gel systems had sufficient mucoadhesive power for vaginal application, and showed a slower release rate than the nanofibers or gel containing the free drug, providing a sustained release; moreover, the nanoparticle-nanofiber formulation showed a 2.5 times higher permeability through excised sheep vaginal mucosa than the nanoparticle-in gel formulation [282].

The CS versatility and ability to interact with negative macromolecules was also exploited to realize innovative vaginal liposomal formulations, where the biopolymer was used both as a membrane component, to increase the physical stability of nanovesicles, and as a coating, to limit the possible burst effect and prolong the in situ residence time, by interacting with the mucosal tissue. Andersen et al. formulated chitosomes, where CS was both embedded in the liposomal membrane and surface-available as a coating layer, able to assure a prolonged release of metronidazole [283]. Mucoadhesive CS-coated liposomes for the vaginal delivery of curcumin [284] or sildenafil citrate [285] were also successfully developed, where the use of the mucoadhesive-permeation enhancer biopolymer allowed the system’s bioadhesiveness and drug bioavailability to increase.

Another type of nanotechnology-based approach for mucosal vaginal delivery consists of the development of mucoadhesive liquid crystalline systems undergoing a fast viscosity increase after application, by the incorporation of water from the vaginal fluid. Curcumin-loaded liquid crystalline systems based on oleic acid/ergosterol as the oily phase, PPG-5-CETETH-20 as the surfactant and 1% CS dispersion as the aqueous phase were suitable for vulvovaginal treatment, allowing the adequate control of the drug release and showing anti-inflammatory and anti-*Candida* activity [286]. More recently, curcumin-loaded liquid crystalline systems containing oleic acid as the oily phase, ethoxylated/propoxylated cetyl alcohol as the surfactant and a CS-polyethyleneimine dispersion as the aqueous phase were proposed as a vaginal mucosal delivery system in the treatment of cervical cancer; in vitro and ex vivo studies proved that the developed formulation had suitable controlled release properties during 12 h and good mucoadhesion and retention ability, and encouraging results were obtained in in vivo angiogenic assays performed with the chick embryo chorioallantoic membrane model [287].

### 4.6. Oral Drug Delivery Systems

The oral route is the most widely used administration route due to its several advantages, such as simple and painlessness self-administration and large patient compliance, safety in use, cost-effectiveness, ease of high-scale production and great formulations’ versatility. However, many drugs have a low oral bioavailability, mainly due to problems of low solubility and/or poor permeability, as well as low stability in the GI (gastro-intestinal) environment, due to possible degradation by the acid pH of the stomach or by the several digestive enzymes present in the GI tract. Moreover, some drugs present narrow absorption windows, or better stability/solubility or higher activity in specific sections of the GI tract.

Many of such issues can be overcome by the development of proper CS-based oral mucosal drug delivery systems, exploiting the several beneficial effects provided by this biopolymer, including wide availability, biocompatibility, biodegradability, absence of toxicity, joined with bioadhesion and permeation enhancer abilities. CS was successfully used to protect the incorporated drug up to the site of action or of absorption, to improve drug solubility, to enhance permeability through the GI mucosa by the opening of tight junctions, to prolong the in situ residence time and promote effective interactions with the mucosal surfaces, due to its mucoadhesion capacity, to control drug release rate; moreover, it offers the possibility to reach specific mucosal targets such as the small intestine and colon for the treatment of topical and systemic pathologies. Several CS derivatives, endowed with further improved properties, such as higher solubility, stability, mucoadhesion and enhancer permeation power, obtained by chemical synthesis [12] or by polyelectrolyte complex (PEC) formation [18] were also investigated.

Furthermore, in addition to its use as a pharmaceutical excipient, CS administration by oral route demonstrated a favorable function in reducing hyperglycemia, dietary fat and cholesterol absorption [288].

CS and its derivatives, alone or in association with a variety of other components or polymers, including as PEC with negatively charged polymers, were employed to formulate a number of both conventional and innovative oral drug delivery systems by virtue of its great formulation versatility. In addition, the peculiar physico-chemical characteristics of CS make it also effective in the development of drug delivery systems targeted to specific sections of the GI tract [289].

The need for the development of increasingly effective gastro-retentive delivery systems, aimed at improving the therapeutic effectiveness of drugs which are better absorbed at gastric pH, or are instable at intestinal pH, or present a narrow absorption window or even are administered for the local treatment of stomach diseases, is always growing.

As illustrated in a recent review [290], most of the more advanced systems realized in the last ten years are based on the use of CS, also exploiting nanotechnological approaches. For example, mucoadhesive nanoparticles of CS, as such [291] or as PEC with alginate [292], loaded, respectively, with ketoconazole or amoxicillin, provided good drug protection from acid degradation and allowed a marked increase in gastric residence time, showing muco-penetrating properties, with a consequent enhanced drug absorption. The success of these CS-based gastro-retentive systems seems also to be related to its ability to increase the gastric delivery of several kinds of drugs [290,293,294].

On the other hand, intestinal delivery can instead be a useful solution for drugs, such as NSAIDs, that can be aggressive for the gastric mucosa or require a specific absorption site in the lower GI tract or for obtaining a time-controlled release. In fact, CS-based oral delivery systems can be designed to reach the lower sections of the GI tract (jejunum, ileum, cecum, colon and rectum) exploiting the pH-responsiveness, enzyme-specific stimuli-release and mucoadhesive properties of CS. The small intestine can also be a target for local treatment, but it is the favorite site for drug absorption, due to its large surface area, being arranged into crypt structures. Several studies demonstrated the protective effect of CS-based drug delivery systems against acidic denaturation and enzymatic degradation, as well as its reversible enhancer effect on the intestinal absorption and its ability to prolong the intestinal residence time due to its intrinsic mucoadhesive characteristics [295].

Tablets realized with CS and carboxymethyl starch loaded with some enzymes were able to protect them against simulated gastric conditions and to control their release in simulated intestinal fluid [296]. Thiolated-CS tablets loaded with insulin showed 80-fold higher mucoadhesiveness and AUC insulin values 21-fold greater than those made with unmodified CS [297].

CS hydrogel beads, obtained by ionotropic-gelation and/or PEC formation with negatively charged polymers, demonstrated their ability to retain their content in an acidic environment and to release it in slightly alkaline conditions [98,298,299,300,301]. CS beads realized by ionic crosslinking with phytic acid, proposed for insulin oral delivery, were able to provide a long-lasting hypoglycemic effect after oral administration in diabetic rats [302]. A CS/tripolyphosphate/graphene oxide hydrogel provided an effective intestinal delivery of sumatriptan succinate, with decreased drug leakage in the stomach [303].

Different types of CS-based pH-sensitive hydrogels were obtained, using suitable CS derivatives, often in combination with alginate to exploit PEC formation, whose drug release rate increased with the increase in pH, making them suitable for intestinal delivery of proteins or their compounds that are susceptible to degradation at gastric pH and enzymatic conditions [126,304,305,306]. Hydrogel beads obtained by association of CS with sodium carboxymethylcellulose, sodium alginate and calcium chloride showed a pH-sensitive behavior, resulting stable for 3 h in simulated gastric fluid and allowing a sustained release of Bacillus subtilis in simulated intestinal fluid for more than 10 h [307].

To further improve the intestinal-targeted release properties, pH- and magnetism-responsive hydrogel beads were prepared, consisting of composite systems made of magnetic nanoparticles mixed with polymers such as carboxymethylcellulose or pectin, and coated with CS [308,309,310]. All of these systems prevented release in the GI tract and showed pH-dependent release profiles of the included model drugs; the application of an external magnetic field can further increase the drug release at the targeted intestinal tract, revealing their potential as multi-responsive delivery systems.

Hydrogels filled with colloidal delivery systems loaded with lipophilic drugs could be an attractive approach to improve the physical and chemical stability of these nanocarriers, enhance their bioavailability and allow a better controlled release of the carried drug. CS hydrogel beads containing liposomes loaded with linseed oil and quercetin showed an improvement of the stability of liposomes in the gastrointestinal tract and slowed the release of the bio-actives, avoiding their rapid release [311].

CS-based nanoengineered systems were investigated by many scientists, who highlighted their advantages as advanced drug delivery systems, being able to protect the entrapped drug from the gastric environment, favor its permeation by different mechanisms (transient opening of tight junctions, reversible increase of trans- and para-cellular permeability, increased residence time through mucus adhesion, possible transport of CS nanoparticles through absorptive endocytosis), control the release rate, and achieve specific intestinal targeting [122,139,312].

CS-tripolyphosphate nanoparticles proved to be able to protect the entrapped enzyme β-galactosilase from the gastric environment and transport it to the brush border membrane of the enterocytes of the small intestine, where the carrier is degraded in a targeted manner by the alkaline phosphatases [313].

Nanoparticles of CS complexed with the mast cell stabilizer cromoglycate were able to increase its intestinal permeability through Caco-2 cells up to 9.3 times compared to the pure drug and to improve up to 1.82 times its relative bioavailability after oral administration in rats [314]. CS nanoparticles loaded with an antigen-cyclodextrin complex orally administered in mice evidenced a 3.6-fold and 1.9-fold increase in intestinal mucosal immune response compared to the free antigen solution or the free antigen-loaded CS nanoparticles, respectively [315].

Nanoparticles obtained by assembling CS together with whey protein and polysaccharides from Ophiopogon japonicus (OJPs) showed improved stability and increased the anti-inflammatory and antioxidant effects of OJPs and allowed an effective protection of the intestinal epithelial barrier against the damages from lipopolysaccharide (LPS)-induced macrophage inflammation, resulting in being useful in the treatment of Inflammatory Bowel Disease [316].

A comparative evaluation of the performance of CS-alginate microspheres and CS- tripolyphosphate nanoparticles as a delivery system for β-carotene showed that both formulations showed satisfying entrapment, increased stability, limited release at gastric pH, and intestine specific diffusion, preserving the drug’s functionality [317].

The CS coating of nanoparticles also demonstrated its efficiency: zein protein nanoparticles loaded with curcumin and berberine and then coated with CS demonstrated higher cytotoxicity against cancer cells than the free drugs, and allowed a greater cellular uptake, apoptosis and inhibitory effect against inflammatory cytokine secretion; moreover, they provided a slow pH-sensitive controlled release, which was higher at pH 5.5 (typical or tumor tissue microenvironment) than at pH 7.4 [318]. On the other hand, magnetic nanoparticles coated with ascorbic acid-modified CS evidenced a good potential as pDNA nanocarrier, proving that the CS coating effectively protected pDNA against the acidic gastric environment and allowed its release when reaching the alkaline intestinal target tissue; additionally, the magnetic nanoparticles could further contribute to the targeted release, by a magnetically driven transfection of the orally administered gene [319].

Colon-targeted delivery of drugs is highly necessary in managing local diseases such as colon cancer, colitis and Inflammatory Bowel Disease. More recently, the colon was also investigated as a site for systemic drug delivery. In fact, even if the colon mucosa is smooth, its surface is less extended than the small intestine and there are no specialized villi, the colon environment can be suitable for peptides and protein delivery, allowing a reduction in the reduce degradation of labile drugs in the duodenum and jejunum conditions, and obtain prolonged and controlled release [320].

The CS mucoadhesive, anti-inflammatory, immunomodulatory and antitumor properties, along with the fact that it cannot be digested in the upper GI tract, while it is specifically degraded by enzymes (chitosanases and others) produced by the bacterial microflora present in the colon [321], make it an ideal candidate in the design of colon-specific delivery systems, where its dissolution in acidic pH is easily avoided by association or coating with other polymers.

CS-based multiparticulate dosage forms, aimed at the colon delivery of drugs in the treatment of Inflammatory Bowel Disease, were recently reviewed [322]. CS-based conventional and unconventional colon-targeted delivery systems were recently reviewed [323,324]; however, the scientific production about such a subject is so broad and articulated, that you can continuously find new CS applications in this field.

Polyelectrolyte complexes (PECs) and CS derivatives can retain their integrity in the upper GI tract and release the entrapped drug in the colon by the enzymatic activity of colonic microflora. A dual pH-/thermo-responsive hydrogel was designed based on a PEC of norbornene-functionalized CS and polyacrylic acid, which was synergized via “click chemistry” using a thermo-sensitive polymeric crosslinker (bistetrazine-poly(N-isopropyl acrylamide); the obtained hydrogel, loaded with a model drug, showed a limited drug release in gastric conditions and an almost complete release in colonic conditions (pH 7.4) [325]. The CS PECs formation was also exploited with several other polymers, such as pectin [326], alginate [327,328,329], thiolated dextran [330] and gellan gum [331], leading to the creation of beads or micro- or nanoparticles aimed for colon delivery.

The formation of CS PECs with pectin was also utilized to obtain enteric-coated tablets, that were able to deliver the drug at an alkaline colonic pH, by a microbial enzyme-dependent degradation mechanism, as proved by ex vivo studies [332].

A novel amphoteric derivative of CS was used for the preparation of enteric-coated tablets that allowed the colon-specific delivery of ronidazole, exploiting a combination of time-, pH-, and enzyme- controlled mechanisms [333].

Monolithic matrices made of CS as such or as PEC with carboxymethyl starch, and loaded with acetaminophen as model drug, showed controlled release profiles, up to 16 h, which could be suitable for intestinal or colonic drug delivery, despite the fact that the tablets were not enterically coated, by virtue of a self-stabilization effect shown by CS during the tablets’ exposure to the gastric acidic medium [334].

The versatile properties of CS can also be exploited in the combined use of different technologies. For example, Eudragit RS nanoparticles, loaded with anticancer drugs, were incorporated into CS/hydroxypropyl methylcellulose microcapsules designed for specific release in the colon, following CS degradation by bacterial enzymes; the enteric-coated biphasic system showed an optimal potential as colon-targeted delivery vehicle for the treatment of colorectal cancer [335].

A multifunctional delivery system was developed for the treatment of colorectal cancer, by co-loading of superparamagnetic nanoparticles and the drug irinotecan into a CS PEC with polyglutamic acid, combining magnetic responsiveness with retention effect and sustained release; the system showed effective internalization by colorectal tumor cells, and promising tumor-targeting capacity, as proved by in vivo biodistribution studies [336].

The use of the 3D printing technology in the production of drug delivery systems is expanding and it appears to be particularly attractive in the nanomedicine field. CS-based polymeric micelles loaded with the anticancer drug camptothecin were inserted into 3D printed systems, that were then coated with an enteric layer to protect the drug and avoid its early release in the upper GI tract; the CS micelles, released at a colonic pH from the 3D printed systems exhibited a significant increase in drug intestinal permeability (P_app_ ≈ 9.10^−6^ cm/s) with respect to the free drug (P_app_ ≈ 5.10^−6^ cm/s), as proved using a 3D intestinal cell-based model [337].

## 5. Conclusions

CS, a cationic polymer obtained by the deacetylation of chitin, is probably the most widely investigated and utilized natural polymer for the development of novel drug delivery systems. Its numerous favorable properties, such as biocompatibility, biodegradability, non-toxicity, mucoadhesiveness and permeation enhancer abilities together with its versatility in forming films, hydrogels, micro- and nanoparticles, make CS a profitable and useful excipient with a very wide and varied range of applications. The main interest in CS-based systems is presently focused on mucosal drug delivery, owing to both the ever-increasing relevance gained by the different mucosal routes as alternative non-invasive delivery pathways, and the unique features of CS, which appear especially suitable to favor delivery and enhance transmucosal drug absorption. Moreover, the ease of chemical modification of CS allows a variety of derivatives to be obtained, able to overcome some possible limitations of the natural polymer, such as its low solubility at neutral pH, and further improve its favorable properties, such as mucoadhesiveness and permeation enhancer.

In this review, after a description of the main properties that make CS an excellent tool for improving the effectiveness of mucosal drug delivery, we presented an updated review of the main and more recent applications of CS and its derivatives in buccal, nasal, ocular, rectal, vaginal and oral delivery, highlighting the specific role played by this biopolymer in facing the challenges presented by each of these different administration sites, and in solving the intrinsic obstacles that each one of them poses to effective drug absorption and therapeutic efficacy.

However, as a final consideration, it should be pointed out that, despite the incredibly wide literature showing the enormous and ever increasing number of examples of successful applications of this fascinating polymer in drug delivery, and especially in mucosal drug delivery, the number of approved drug products with CS as excipient is still limited. The reasons for this are mainly attributable to a lack of an adequate characterization (about purity, deacetylation degree, deacetylation pattern, molecular weight, etc.) of CS and its derivatives, which is considered, at least in part, responsible for some contradictory results reported in literature about the effects of this polymer.

Surely, the very promising results obtained proved the great potential of CS in applications for the different routes of mucosal drug delivery. Therefore, it is strongly desirable that a proper standardization of CS and its derivatives will soon be achieved, so as to enhance the possibility of its commercialization and allow a more effective exploitation of its several beneficial properties.

## Figures and Tables

**Figure 1 marinedrugs-20-00335-f001:**
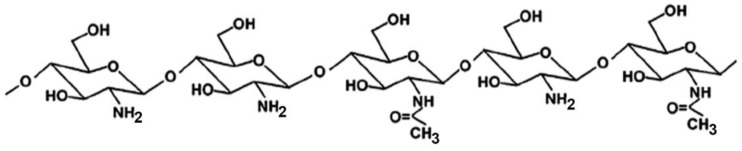
Representative structure of partially N-acetylated chitosan.

**Figure 2 marinedrugs-20-00335-f002:**
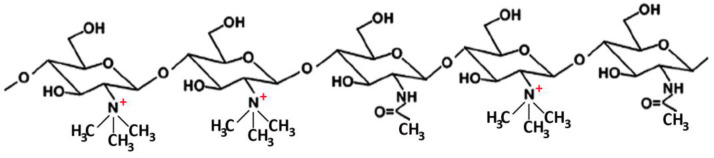
Representative schematic structure of N, N, N-trimethyl chitosan.

**Figure 3 marinedrugs-20-00335-f003:**
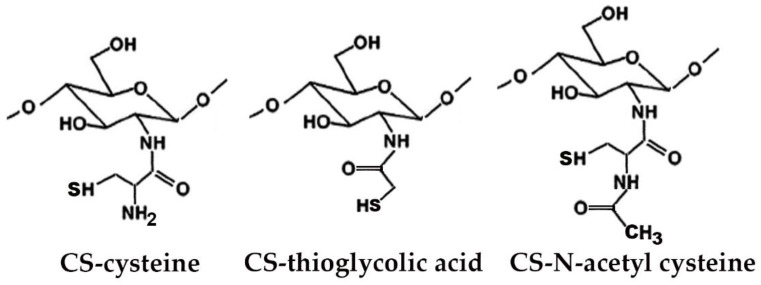
Representative schematic structures of some examples of thiolated chitosan (CS) derivatives.

**Figure 4 marinedrugs-20-00335-f004:**
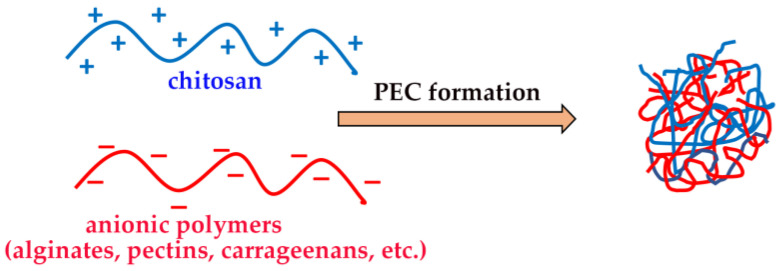
Chitosan-based polyelectrolyte complexes (PECs).

**Figure 5 marinedrugs-20-00335-f005:**
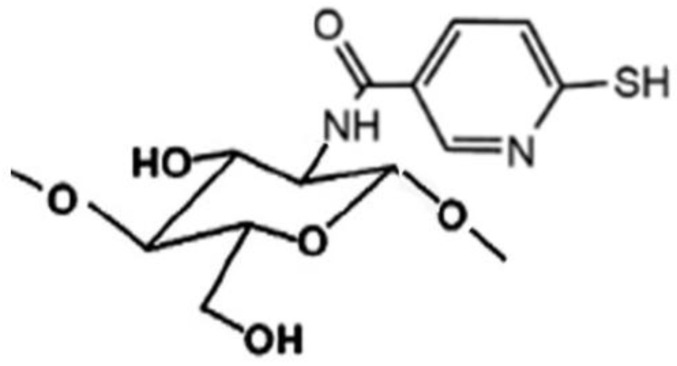
Representative structure of sulfhydryl-linked chitosan-mercaptonicotinic acid.

**Figure 6 marinedrugs-20-00335-f006:**
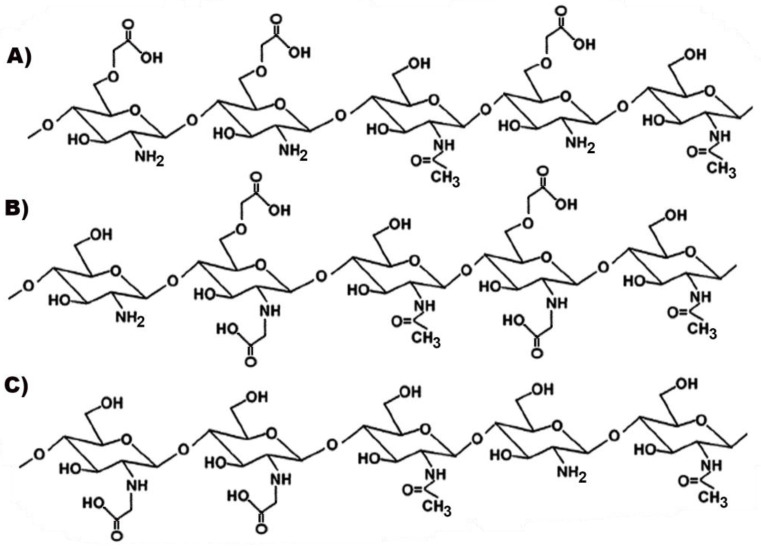
Representative structures of different types of carboxymethylated chitosan (CS): (**A**) O-carboxymethyl-CS; (**B**) N, O-carboxymethyl-CS; (**C**) N-carboxymethyl-CS.

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
