# Peer review of "Multiple Roles of Chitosan in Mucosal Drug Delivery: An Updated Review"

_marinedrugs, 2022, doi:10.3390/md20050335_

Round 1
Reviewer 1 Report
Title: Multiple roles of chitosan in mucosal drug delivery: un up- 2 dated review
I would like to thank the authors for their efforts in making this review available to the public this review that aims to provide an updated view of the applications of chitosan and its derivatives in novel formulations intended for different forms of mucosal drug delivery.
Throughout the text the expression in fact is repeated a lot, I recommend to the authors a general revision of the text in the English language and to change this expression for some other one.
Title and abstract
The title is correct and descriptive. The ideas in the abstract are clear and the topic is well explained. I do not have any considerations in this part.
- Introduction
In general, this section is clear, but I also suggest a slight revision and a correct these suggestions:
L28: Why the via oral is the form favorite by patients?
L87-L90: I don´t understand this paragraph. Please I recommend the authors rewrite it.
- Properties of CS making it a good candidate for mucosal drug delivery
I think that this part is well structured and correct. However, there are some minor mistakes, listed below:
L106: In the title of the section there is an extra period.
L118: “as shown in a recent review” I recommend the authors to mention in the text some examples of these beneficial properties that it possesses or talk about the recent review that is also mentioned in the text. It is necessary to give a little more information.
L178 – 179: This paragraph has no bibliography?
- Limitations of chitosan
In my opinion, I consider this section to be correct and well organized, but I would recommend the authors in the line 235: “to” is not necessary in this section and “use” I recommend the author adding an article.
- Chitosan-based mucosal drug delivery systems
In general, this section well expressed. However, I will suggest some modifications or corrections to improve the document:
L260: “in fact” is not necessary in this sentence. Consider removing it.
L310: Please add a letter “s” to the word “sensation”
L342: I recommend the author change the words “with respect to” for “concerning for to”.
L373: “in vivo” is necessary to write in italic.
L478: Please add “a” before the word “maximum”
L479: “damages” is better in singular “damage”
L513: “In fact” is not necessary.
L514: The phrase following the intransitive verb “resulted” seems to be missing a preposition. Consider adding one as for example “resulted in” or “resulted from”.
L527: Please change “that” for “which”
L532: Please consider add an article in this expression “nose-to-brain”.
L537: Please consider add an article in this sentence “transport drug”
L884: “In situ” must in italic. All word “in silico, in vitro, in vivo, in situ” Should be in italics. Review this throughout the document.
L891: “an effective” the word “an” is not necessary.
L975: Please change “favourable” for “favorable”.
- Conclusions
The conclusions are very clear and well written, but in the line1384: The verb obtain is usually in the gerund form when following the word allowed. Consider replacing it whit the -ing form. In the line 1394: please change “oppose” for “opposes”.
Tables and Figures
Figure 1: Please I recommend the author to improve the image quality because it is blurred. Explained something about image in the footer.
In this review does not appear any table.
Wouldn't it be interesting to put some more images or a table?
References
I consider that is correct. I do not have any consideration in this section.
Final Remarks
For improvement, the manuscript should be revised according to the above suggestions and those of other reviewers. In my honest opinion, I suggest a minor revision of the article. The authors have done work that provides interesting results.
Reviewer 2 Report
The paper is well-written and contains a comprehensive study of the subject. Just some minor points, which have been commented on in the attached file should be addressed before final acceptance.
It is also more useful if the authors could add some schemes to the manuscript, e.g. showing some modified structures of Cs in each section.
